# BayesPCN: A Continually Learnable Predictive Coding Associative Memory

**Jason Yoo**
Department of Computer Science
University of British Columbia
Vancouver, Canada
jasony97@cs.ubc.ca

**Frank Wood**
Department of Computer Science
University of British Columbia
Vancouver, Canada
fwood@cs.ubc.ca

## Abstract

Associative memory plays an important role in human intelligence and its mechanisms have been linked to attention in machine learning. While the machine learning community's interest in associative memories has recently been rekindled, most work has focused on memory recall ($read$) over memory learning ($write$). In this paper, we present BayesPCN, a hierarchical associative memory capable of performing continual one-shot memory writes without meta-learning. Moreover, BayesPCN is able to gradually forget past observations ($forget$) to free its memory. Experiments show that BayesPCN can recall corrupted i.i.d. high-dimensional data observed hundreds to a thousand "timesteps" ago without a large drop in recall ability compared to the state-of-the-art offline-learned parametric memory models.

## 1 Introduction

One of the hallmarks of biological intelligence is the ability to robustly recall learned associations. Biological agents are able to associate stimuli of various modalities and rely on their memories to perform everyday tasks from navigation to entity recognition. Yet these agents are not only good at recall but can quickly and continually form new memories from as little as a single experience. How do biological agents achieve this feat while mitigating catastrophic forgetting [French, 1999]? And how can we build neural networks that demonstrate these properties? This paper addresses these questions, focusing on fast, continual learning of neural artifacts that perform robust recall of i.i.d. high-dimensional observations.

Associative memory models have been of interest in machine learning for decades [Steinbuch, 1961, Hopfield, 1982]. They have been used to solve a wide range of problems from sequence processing [Schmidhuber, 1992, Graves et al., 2014, Schlag et al., 2021] to pattern detection [Widrich et al., 2020, 2021, Seidl et al., 2021]. Their study has allowed us to build models that replicate aspects of biological intelligence [Kanerva, 1988, Hawkins and Blakeslee, 2004] and provided novel perspectives on machine learning toolsets like dot-product attention, transformers, and linear layers of deep neural networks [Ramsauer et al., 2020, Bricken and Pehlevan, 2021, Irie et al., 2022].

Our main aim is to recall exact memories from a neural network that can rapidly and continually learn from a stream of data. Towards this end we introduce BayesPCN, a Bayesian take on the generative predictive coding network (GPCN) [Salvatori et al., 2021] that combines predictive coding and locally conjugate Bayesian updates[1]. There are currently a number of associative memory systems that are continually learnable [Hopfield, 1982, Kanerva, 1988, Hawkins and George, 2007, Ramsauer et al., 2020, Sharma et al., 2022]. However, to our knowledge, BayesPCN is the first parametric associative memory model to continually learn hundreds of high-dimensional observations while maintaining

---

[1]Code is available at https://github.com/plai-group/bayes-pcn

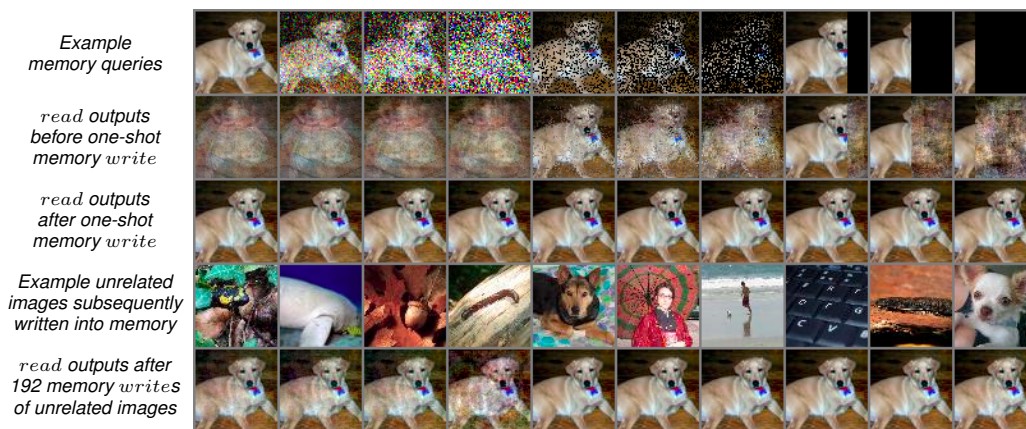

*Example memory queries*

*read outputs before one-shot memory write*

*read outputs after one-shot memory write*

*Example unrelated images subsequently written into memory*

*read outputs after 192 memory writes of unrelated images*

Figure 1: BayesPCN recall before and after the one-shot $write$ of the top left image into memory. The first row contains example memory queries, the second row contains the memory's $read$ outputs before $write$, the third row contains the memory's $read$ outputs immediately after $write$, the fourth row contains sample images written into memory after the top left image, and the last row contains the memory's $read$ outputs after 192 additional datapoints have been sequentially written into memory. The model has 3 hidden layers of width 1024. The four leftmost columns illustrate auto-associative recall while the remaining columns illustrate hetero-associative recall.

good recall performance on inputs with nontrivial amounts of corruption. In addition, it is also the first hierarchical memory which after repeated forgetting recovers its original memory state.

## 2 Background

### 2.1 Auto-Associative and Hetero-Associative Memory

Before we begin let us review two useful concepts: auto-associative memory and hetero-associative memory. The former, auto-associative memory, refers to a memory system that is content addressable but for which the query may be a corrupted version of the original data item. One can think about auto-associative memory systems as a kind of approximate nearest neighbor lookup mechanism. Hetero-associative memory systems are key-value stores, meaning that the recalled "value" is potentially of a different type to the "key." Crucially, hetero-associative memory systems can, in large part, be implemented by auto-associative memory systems by forming a "joint" object to memorize by combining the key and value into one object. Hetero-associative queries can then be processed simply by looking up the joint object "corrupted" by discarding the value.

### 2.2 Generative Predictive Coding Network

A generative predictive coding network (GPCN) [Salvatori et al., 2021] is an energy-based associative memory model that has set the state-of-the-art on a number of image associative recall tasks. As illustrated in Figure 2, GPCN takes the form of a fully connected neural network whose bottom layer activations characterize the sensory inputs and higher layer activations characterize the internal representations of the sensory inputs. GPCN's $read$ operation (recall) initializes the bottom layer activations to the input query vector and iteratively denoises the query until a stored value is retrieved by minimizing its energy function w.r.t. some or all of its neuron activations. GPCN's $write$ operation (learning) adapts the network parameters to shape the energy function such that the stored datapoints become the local minima of the energy function.

Before describing GPCN's $read$ and $write$ mechanisms in more detail, we introduce the notation we use in this paper. Let $M$ be a GPCN with $L + 1$ layers of neurons where $0$ and $L$ denotes the bottom and the top layer index respectively. Then, let $d_l$ be the width of the $l$-th layer, $x^l \in \mathbb{R}^{d_l}$ be the row vector denoting the $l$-th layer's pre-activation function neuron values, and $f^l : \mathbb{R}^{d_l} \to \mathbb{R}^{d_l}$ be the $l$-th layer's activation function, for instance ReLU. In addition, let $W^l \in \mathbb{R}^{d_{l+1} \times d_l}$ where

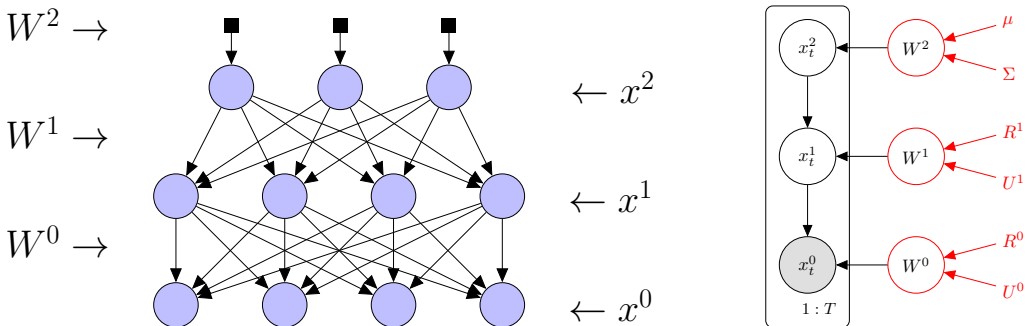

Figure 2: **Left**: A sample 3-layer GPCN/BayesPCN architecture. Circular nodes denote neurons, arrows denote synaptic weights, $x^0$ denotes the sensory neuron activations, $x^1, x^2$ denote the upper layer hidden neurons activations, $W^0, W^1$ denote the synaptic weights between network layers, and $W^2$ denotes the bias term that influences the top layer neuron activities. **Right**: graphical model for the memory model to the left with $T$ i.i.d. sensory observations $x^0_{1:T}$. GPCN's graphical model is black while BayesPCN's graphical model adds the red components. GPCN treats the synaptic weights as constants while BayesPCN treats them as normally distributed random variables.

$0 \leq l < L$ be the synaptic weight matrix between $l+1$-th and $l$-th layers and $W^L \in \mathbb{R}^{d_L}$ be the "bias" row vector that influences the dynamics of the top layer neuron activities $x^L$. One can intuitively think of the model parameters $W^{0:L}$ as the memory content and the activations $x^{0:L}$ as the read state of the memory.

GPCN's $read$ and $write$ are facilitated by predictive coding (PC) [Rao and Ballard, 1999], a biologically plausible learning rule that can approximate backpropagation [Millidge et al., 2020]. PC's energy function is the sum of prediction errors across all network layers. For GPCN, the choice of prediction error is the squared Euclidean distance between the actual layer activations and the prediction of the same layer activations made by their immediate upper layer activations. This leads to the following energy function.

$$E(x^{0:L}, W^{0:L}) = \frac{1}{2}||x^L - W^L||^2 + \frac{1}{2}\sum_{l=0}^{L-1}||x^l - f^{l+1}(x^{l+1})W^l||^2 \tag{1}$$

GPCN's $read$ performs gradient descent on the energy w.r.t. some or all of $x^0$ and $x^{1:L}$ after setting the initial values of $x^0$ to the memory query $\bar{x}^0$ and fixing the values of $W^{0:L}$. GPCN's $write$ obtains the $read$ output $x^{0:L}_{1:T}$ for each of $T$ i.i.d. datapoints to store $x^0_{1:T}$, takes a gradient step on the energy w.r.t. $W^{0:L}$ while fixing $x^{0:L}_{1:T}$, and repeats until convergence.

We now take a probabilistic perspective on GPCN. One can view GPCN as a hierarchical generative model where all neuron activations are modeled by normally distributed random variables and the link function between the subsequent layers is a layer-specific nonlinearity followed by a linear transform. Then, GPCN's $read$ - the minimization of Equation 1 - is equivalent to maximizing the log density in Equation 2 w.r.t. $x^0$ and $x^{1:L}$.

$$\log p(x^{0:L}|W^{0:L}) = \log \mathcal{N}(x^L; W^L, I) + \sum_{l=0}^{L-1}\log \mathcal{N}(x^l; f^{l+1}(x^{l+1})W^l, I) \tag{2}$$

In other words, GPCN's $read$ performs iterated conditional modes [Besag, 1986] on the hierarchical generative model in the neuron activation space (refer to Algorithms 3 and 4). In addition, GPCN's $write$ can be viewed as an approximate maximization of the lower bound on $\log p(x^0_{1:T}|W^{0:L})$ w.r.t. $W^{0:L}$ [Millidge et al., 2021]. This follows from the fact that GPCN learns via PC.

While GPCN possesses impressive associative recall capabilities, the learning rule proposed in Salvatori et al. [2021] is not amenable to continual learning. Due to its lack of safeguards against catastrophic forgetting, continually learning GPCN leads to a forgetting of information associated with older observations or requires a costly offline model refitting on all observations whenever a new datapoint is observed. BayesPCN's foremost aim is to address this limitation while preserving GPCN's $read$ capabilities as much as possible.

## 3 Memory Model

BayesPCN is structurally identical to GPCN, but unlike GPCN the network parameters $W^{0:L}$ are treated as latent random variables. This allows BayesPCN's *write* operation to be cast as (approximate) posterior inference over $W^{0:L}$ given new observations in a way similar to [Wu et al., 2018a]. In other words, storing $T$ i.i.d. datapoints $x_{1:T}^0$ into memory is equivalent to estimating $p(W^{0:L}|x_{1:T}^0)$. Figure 2 illustrates the graphical model for BayesPCN.

We now review the probabilistic structure of BayesPCN. We assume that, prior to any observations, the random variables $x_t^{0:L}$ and $W^{0:L}$ (where $t$ is the datapoint index) are distributed as follows

$$p(W^l) = \begin{cases} \mathcal{MN}(W^l; R^l, \sigma_W^2 I, I) & \text{if } 0 \leq l < L \\ \mathcal{N}(W^L; \mu, \sigma_W^2 I) & \text{if } l = L \end{cases} \tag{3}$$

$$p(x_t^l|x_t^{l+1}, W^l) = \begin{cases} \mathcal{N}(x_t^l; f^{l+1}(x_t^{l+1})W^l, \sigma_x^2 I) & \text{if } 0 \leq l < L \\ \mathcal{N}(x_t^L; W^L, \sigma_x^2 I) & \text{if } l = L \end{cases} \tag{4}$$

where $\mathcal{N}(\mu, \Sigma)$ and $\mathcal{MN}(R, U, V)$ denotes normal and matrix normal distribution [Gupta and Nagar, 2018] respectively. We remind the readers that the matrix normal distribution is a normal distribution for matrix-variate data that compactly specifies the matrix elements' means, row-wise covariances, and column-wise covariances. The joint distribution over all random variables when there are $T$ sensory observations is, due to the conditional independence of $x_t^{0:L}, t \in \{1, ..., T\}$ given $W^{0:L}$,

$$p(x_{1:T}^{0:L}, W^{0:L}) = \left[\prod_{t=1}^{T} p(x_t^{0:L}|W^{0:L})\right] p(W^{0:L}) = \left[\prod_{t=1}^{T}\prod_{l=0}^{L} p(x^l|x^{l+1}, W^l)\right]\prod_{l=0}^{L} p(W^l) \tag{5}$$

We note that BayesPCN's structure leads to two desirable properties that become useful when approximating $p(W^{0:L}|x_{1:T}^0)$ during the *write* operation. First, by exploiting the linear Gaussian marginalization formula and conditional independence we can analytically compute the marginal distribution over all activation vectors $x_t^{0:L}$.

$$p(x_t^{0:L}) = \int \prod_{l=0}^{L} p(x_t^l|x_t^{l+1}, W^l)p(W^l)dW^{0:L} = \prod_{l=0}^{L} p(x_t^l|x_t^{l+1}) \tag{6}$$

$$p(x_t^l|x_t^{l+1}) = \begin{cases} \mathcal{N}(x_t^l; f^{l+1}(x_t^{l+1})R^l, (\sigma_x^2 + f^{l+1}(x_t^{l+1})U^l f^{l+1}(x_t^{l+1})^\top)I) & \text{if } 0 \leq l < L \\ \mathcal{N}(x_t^L; \mu, (\sigma_x^2 + \sigma_W^2)I) & \text{if } l = L \end{cases} \tag{7}$$

We stress that neither $p(x_t^0)$ nor $p(x_t^0|W^{0:L})$ has an analytic form due to the presence of nonlinearities. In addition, if all activation vectors $x_t^{0:L}$ are observed, the posterior over the model parameters becomes normally distributed due to the linear Gaussian posterior formula and the fact that $W^{0:L}$ are conditionally independent given $x_t^{0:L}$ (the analytical posterior formulae for Equation 9 are in Appendix C).

$$p(W^{0:L}|x_t^{0:L}) = \frac{1}{Z}\prod_{l=0}^{L} p(x_t^l|x_t^{l+1}, W^l)p(W^l) = \prod_{l=0}^{L} p(W^l|x_t^l, x_t^{l+1}) \tag{8}$$

$$p(W^l|x_t^l, x_t^{l+1}) = \begin{cases} \mathcal{MN}(W^l; R^{l*}, U^{l*}, I) & \text{if } 0 \leq l < L \\ \mathcal{N}(W^L; \mu^*, \Sigma^*) & \text{if } l = L \end{cases} \tag{9}$$

We use $R^{l*}, U^{l*}, \mu^*, \Sigma^*$ to denote the sufficient statistics of the layer-wise network weight posterior distributions given $x_t^l, x_t^{l+1}$.

## 4 Memory Operations

This section contains the memory operations that BayesPCN supports: *read*, *write*, and *forget*. We use the shorthand $\mathbf{h} = x^{1:L}$ to denote all hidden layer neuron activations and $\mathbf{W} = W^{0:L}$ to denote all network parameters.

## 4.1 Read

The $read$ operation takes the corrupted query vector $\bar{x}^0$ and retrieves the memory's best guess of the associated original datapoint $x^{0*}$. As in GPCN, this is done by hill climbing the memory's log density $\log p(x^0, \mathbf{h}|x_{1:T}^0)$ w.r.t. the neuron activations $x^0, \mathbf{h}$. To clarify, $x^0, \mathbf{h}$ is the current memory read state and is not directly related to the past observations $x_{1:T}^0$. BayesPCN's auto-associative recall performs iterated conditional modes on $\log p(x^0, \mathbf{h}|x_{1:T}^0)$ w.r.t. $x^0, \mathbf{h}$ and BayesPCN's hetero-associative recall performs gradient ascent on $\log p(x^0, \mathbf{h}|x_{1:T}^0)$ w.r.t. $x^0, \mathbf{h}$ while keeping the known values of $x^0$ fixed. Algorithm 3 and Algorithm 4 in Appendix A, the recall procedures taken from Salvatori et al. [2021] and rewritten from the probabilistic perspective, describe this in more detail.

## 4.2 Write

The $write$ operations aims to take the previous timestep posterior over the model parameters $p(\mathbf{W}|x_{1:t-1}^0)$ and the current timestep observation $x_t^0$ to return the current timestep posterior over the model parameters $p(\mathbf{W}|x_{1:t}^0)$. Unfortunately, the analytical form of $p(\mathbf{W}|x_{1:t}^0)$ is unavailable due to the presence of nonlinearities in BayesPCN so one must resort to approximate posterior inference.

We propose a sequential importance sampling posterior inference algorithm for estimating $p(\mathbf{W}|x_{1:t}^0)$ where the particles are high-dimensional Gaussian distributions. Let $N \in \mathbb{N}$ be the number of sequential importance sampling particles and $n \in \{1, ..., N\}$ be the particle index. The $t$-th timestep posterior estimate $\hat{p}(\mathbf{W}|x_{1:t}^0)$ is recursively represented as a weighted Gaussian mixture of the form $\hat{p}(\mathbf{W}|x_{1:t}^0) = \sum_{n=1}^{N} \omega^{(n)} p^{(n)}(\mathbf{W}|x_{1:t}^0, \mathbf{h}_{1:t}^{(n)})$ where $\omega^{(n)}$ is the $n$-th particle's importance weight and $p^{(n)}(\mathbf{W}|x_{1:t-1}^0, \mathbf{h}_{1:t-1}^{(n)})$ is the $n$-th particle's activation conditioned posterior described in Equation 8. We note that each $p^{(n)}(\mathbf{W}|x_{1:t-1}^0, \mathbf{h}_{1:t-1}^{(n)})$ is efficiently characterised by the sufficient statistics $\mu^{(n)}, \Sigma^{(n)}, R^{0:L-1,(n)}, U^{0:L-1,(n)}$. Algorithm 1 describes the $write$ operation in detail (see Appendix B for its justification).

---

**Algorithm 1** Memory Write

---

1: **Input**: Previous network weight distribution $\hat{p}(\mathbf{W}|x_{1:t-1}^0)$, current timestep observation $x_t^0$
2: **Output**: Updated network weight distribution $\hat{p}(\mathbf{W}|x_{1:t}^0)$

---

3: **for** each model $p^{(n)}(\mathbf{W}|x_{1:t-1}^0, \mathbf{h}_{1:t-1}^{(n)})$ and weight $\omega_{t-1}^{(n)}$ in $\hat{p}(\mathbf{W}|x_{1:t-1}^0)$ **do**
4:     Fit the parameters of the current timestep variational distribution:
5:        $\phi = \operatorname{argmin}_\phi D_{KL}\left( q_\phi^{(n)}(\mathbf{h}_t) \,||\, \hat{p}^{(n)}(x_t^0, \mathbf{h}_t|x_{1:t-1}^0) \right)$
6:     Sample $\mathbf{h}_t^{(n)}$: $\mathbf{h}_t^{(n)} \sim q_\phi^{(n)}(\mathbf{h}_t)$
7:     Update the model via linear Gaussian update conditioned on $x_t^0$ and $\mathbf{h}_t^{(n)}$:
8:        $p^{(n)}(\mathbf{W}|x_{1:t}^0, \mathbf{h}_{1:t}^{(n)}) \propto p^{(n)}(\mathbf{W}|x_{1:t-1}^0, \mathbf{h}_{1:t-1}^{(n)}) p^{(n)}(x_t^0, \mathbf{h}_t^{(n)}|\mathbf{W})$
9:     Compute the unnormalized particle weight: $\hat{\omega}_t^{(n)} = \omega_{t-1}^{(n)} \frac{p^{(n)}(x_t^0, \mathbf{h}_t^{(n)}|x_{1:t-1}^0)}{q_\phi^{(n)}(\mathbf{h}_t^{(n)})}$
10: **end for**
11: Set $\hat{p}(\mathbf{W}|x_{1:t}^0) = \sum_{n=1}^{N} \frac{\hat{\omega}_t^{(n)}}{\sum_{n'=1}^{N} \hat{\omega}_t^{(n')}} p^{(n)}(\mathbf{W}|x_{1:t}^0, \mathbf{h}_{1:t}^{(n)})$

---

While $q^{(n)}(\mathbf{h}_t)$ in principle can be any distribution, BayesPCN's $write$ sets it to $q^{(n)}(\mathbf{h}_t) = \delta\left(\operatorname{argmax}_{\mathbf{h}_t} \log p(x_t^0, \mathbf{h}_t|x_t^{0:L})\right)$ as is standard in predictive coding [Millidge et al., 2021]. In addition, we define the BayesPCN parameter prior (of an "empty" memory) as $\hat{p}(\mathbf{W}) = \sum_{n=1}^{N} \frac{1}{N} p^{(n)}(\mathbf{W})$ where $\mu^{(n)}, R^{0:L-1,(n)}$ are initialized using Kaiming initialization. We chose this prior over a non-mixture prior to obtain a multi-particle posterior approximation despite using a Dirac variational distribution.

We now direct our attention to obtaining $\log p^{(n)}(x_t^0, \mathbf{h}|x_{1:t-1}^0)$ used in the $read$ operations from $\hat{p}(\mathbf{W}|x_{1:t}^0)$. Using Equation 6, we obtain the approximate posterior predictive density

$$\log p(x_t^0, \mathbf{h}|x_{1:t-1}^0) = \log \sum_{n=1}^{N} \omega^{(n)} \int p(x_t^0, \mathbf{h}_t|\mathbf{W})\hat{p}^{(n)}(\mathbf{W}|x_{1:t-1}^0)d\mathbf{W} \qquad (10)$$

$$= \log \sum_{n=1}^{N} \omega^{(n)} p^{(n)}(x_t^0, \mathbf{h}_t|x_{1:t-1}^0) \qquad (11)$$

BayesPCN's $write$ can be seen as a special case of sequential imputation [Kong et al., 1994], an online parameter inference algorithm that represents the parameter posterior conditioned on partial observations as a mixture of parameter posteriors conditioned on complete observations. In addition, whereas the parameter update proposed by Salvatori et al. [2021] is doing local gradient descent for each $W^l$, BayesPCN's parameter update is doing local conjugate normal-normal Bayesian update for each $W^l$. Hence, like GPCN, BayesPCN's synaptic weight update only utilizes local information. Lastly, while exact Bayesian updates are invariant to the order of observations, because we are using predictive coding to fit the variational distribution BayesPCN's $write$ is technically not invariant to the order of observations.

### 4.3 Forget

We also propose a diffusion-based forgetting mechanism for BayesPCN. As more datapoints are written into a BayesPCN model, the model's belief over the true parameter location $R^{1:L-1}, \mu$ shifts and the uncertainty over its belief $U^{1:L-1}, \Sigma$ decreases. This has the effect of making the model's parameters less adaptive to new datapoints and typically magnifying $R^{1:L-1}, \mu$'s norms as the prior's contribution weakens. One can interpret this as the memory being overloaded. The $forget$ operation counteracts this phenomenon by nudging the model's belief over $R^{1:L-1}, \mu$ toward the prior (empty memory state) and increasing its uncertainty $U^{1:L-1}, \Sigma$ proportional to the forget strength parameter $\beta$. Algorithm 2 describes this in detail (the analytical diffusion formulae are in Appendix D).

---

**Algorithm 2** Memory Forget

---

1: **Input**: Previous network weight distribution $\hat{p}(\mathbf{W}|x_{1:t}^0)$, forget strength $\beta \in [0, 1]$
2: **Output**: Updated network weight distribution $\hat{p}(\mathbf{W}'|x_{1:t}^0)$

---

3: **for** each model $p^{(n)}(\mathbf{W}|x_{1:t}^0, \mathbf{h}_{1:t}^{(n)})$ in $\hat{p}(\mathbf{W}|x_{1:t}^0)$ **do**
4:     **for** each layer weight distribution $p^{(n)}(W^l|x_{1:t}^{(n),l}, x_{1:t}^{(n),l+1})$ in $p^{(n)}(\mathbf{W}|x_{1:t}^0, \mathbf{h}_{1:t}^{(n)})$ **do**
5:         Obtain the one-step diffused layer weight distribution $p^{(n)}(W^{l'}|x_{1:t}^{(n),l}, x_{1:t}^{(n),l+1})$:
6:         $p^{(n)}(W^{l'}|x_{1:t}^{(n),l}, x_{1:t}^{(n),l+1}) = \int p(W^{l'}|W^l, \beta)p^{(n)}(W^l|x_{1:t}^{(n),l}, x_{1:t}^{(n),l+1})dW^l$
7:     **end for**
8:     Set $p^{(n)}(\mathbf{W}'|x_{1:t}^0, \mathbf{h}_{1:t}^{(n)}) = \prod_{l=0}^{L} p^{(n)}(W^{l'}|x_{1:t}^{(n),l}, x_{1:t}^{(n),l+1})$
9: **end for**
10: Set $\hat{p}(\mathbf{W}'|x_{1:t}^0) = \sum_{n=1}^{N} \omega^{(n)} p^{(n)}(\mathbf{W}'|x_{1:t}^0, \mathbf{h}_{1:t}^{(n)})$

---

If applied an infinite number of times, $forget$ reverts the posterior over the parameters $\hat{p}(\mathbf{W}|x_{1:t}^0)$ to the prior $\hat{p}(\mathbf{W})$ due to the property of diffusion if the particle weights are uniform. In practice, one would apply $forget$ periodically or when the memory starts to become overloaded from too much data and stops being performant. Because $forget$ erases stored information for all of datapoints somewhat evenly, the datapoints from the distant past that $forget$ was applied to more would be, as the name suggests, more forgotten than the recent datapoints that $forget$ was applied to less.

## 5 Related Work

We briefly review the most relevant work to BayesPCN, starting with the predictive coding networks. As covered in Section 2, GPCN [Salvatori et al., 2021] is the model that BayesPCN is built upon. Self-taught associative memory [Dovrolis, 2018, Smith et al., 2021] is a hierarchical predictive coding memory that performs unsupervised continual learning using online clustering modules as

building blocks, unlike BayesPCN's building blocks of artificial neurons. The sequential neural coding network [Ororbia et al., 2019, 2018], while not explicitly used as an associative memory, is also a continually learned predictive coding network similar in structure to GPCN that combats catastrophic forgetting via task-dependent activation sparsity. BayesPCN on the other hand combats catastrophic forgetting by incorporating uncertainty over the synaptic weights.

Moving on to general attractor-based memory models, Hopfield networks [Hopfield, 1982] and their modern variants [Krotov and Hopfield, 2016, Ramsauer et al., 2020, Krotov and Hopfield, 2020, Krotov, 2021, Millidge et al., 2022] are biologically motivated auto-associative memories that define the neuron activation dynamics via Lagrangian functions. Out of these models only Krotov [2021]'s model is hierarchical, but theirs is not continually learned. BayesPCN generalizes the modern Hopfield network and fits into the universal Hopfield network's similarity-separation-projection paradigm for associative memories (refer to Appendix E). Sparse Distributed Memory (SDM) [Kanerva, 1988] is a non-hierarchical memory that exploits high-dimensionality to perform auto-associative and hetero-associative recall. Hierarchical Temporal Memory (HTM) [Hawkins and George, 2007, Cui et al., 2016] is a biologically constrained hierarchical memory that continually learns time-based patterns with sparse representations. HTM does not employ predictive coding and to our knowledge has not yet been tested on high-dimensional data.

Relative to generative modeling and memory research, our work has connections to the Kanerva machine's memory $write$ as inference approach [Wu et al., 2018a,b] and the one-shot memory write from Bartunov et al. [2019], Pham et al. [2022]. These papers' models employ meta-learning unlike aforementioned work. We note that the Kanerva machine also applies conjugate normal-normal Bayesian update to adapt its top level memory matrix once the model is meta-learned via end-to-end gradient descent. On the other hand, BayesPCN applies conjugate normal-normal Bayesian update to all of its weight matrices and this as of itself constitutes learning.

## 6 Experiments

This section assesses BayesPCN's recall capabilities on image recovery tasks and examines the effect of $forget$ on BayesPCN's behaviour.

We used CIFAR10 [Krizhevsky et al., 2009] and Tiny ImageNet [Le and Yang, 2015] datasets for all experiments, whose images are of size $3 \times 32 \times 32$ and $3 \times 64 \times 64$ respectively. During the $write$ phase of the experiments, BayesPCN models were given one image to store into their memories per timestep for up to 1024 images. Then, during the $read$ phase of the experiments, we recorded the pixel-wise mean squared error (MSE) between the $write$ phase images and the models' $read$ outputs given the corrupted $write$ phase images as inputs. For reference, the pixel-wise MSE of $0.01$ is distinguishable to human eyes on close inspection of the images.

BayesPCN models were compared against identity models (models that return their inputs as outputs), modern Hopfield networks (MHN) [Ramsauer et al., 2020], and vanilla GPCNs [Salvatori et al., 2021] all of which were offline learned. We note that GPCNs have set the state-of-the-art on CIFAR10 and TinyImageNet image recall tasks. BayesPCN models were also compared against online GPCNs, GPCNs that naively applied their offline memory $write$ to sequentially arriving observations and memorized one image per timestep for up to 1024 images. We used our own autodiff implementation of MHNs based on Appendix E. See Appendix F for additional experiment details.

### 6.1 Auto-Associative and Hetero-Associative Read

For the auto-associative recall experiments, the memory inputs at the $read$ phase were the $write$ phase images corrupted by pixel-wise white noise with standard deviation $0.2$. For the hetero-associative recall experiments, we tested two types of training image corruption schemes. The first type, dropout, randomly blacked out 25% of the image pixels. The second type, mask, blacked out 25% of the rightmost image pixels. Image recovery under these settings is hetero-associative recall because one can view the non-blacked out pixels as the "key" and the blacked out pixels as the "value". We experimented with 4-layer BayesPCN models that have hidden layer sizes of 256, 512, and 1024, particle counts of 1 and 4, and activation functions ReLU and GELU. We searched over the same hyperparameters for offline and online 4-layer GPCNs aside from the particle count. Lastly, we

| White Noise CIFAR10 MSE | | | | |
|---|---|---|---|---|
| Sequence Length | 128 | 256 | 512 | 1024 |
| Identity | 0.1596 | 0.1600 | 0.1600 | 0.1600 |
| MHN | 0.0000 | 0.0000 | 0.0000 | 0.0000 |
| GPCN (Offline) | 0.0028 | 0.0046 | 0.0073 | 0.0121 |
| GPCN (Online) | 0.0103 | 0.0150 | 0.0191 | 0.0210 |
| BayesPCN | 0.0017 | 0.0085 | 0.0146 | 0.0337 |
| BayesPCN ($forget$) | 0.0064 | 0.0102 | 0.0145 | 0.0188 |

| White Noise Tiny ImageNet MSE | | | | |
|---|---|---|---|---|
| Sequence Length | 128 | 256 | 512 | 1024 |
| Identity | 0.1600 | 0.1602 | 0.1601 | 0.1600 |
| MHN | 0.0000 | 0.0000 | 0.0000 | 0.0000 |
| GPCN (Offline) | 0.0005 | 0.0010 | 0.0018 | 0.0067 |
| GPCN (Online) | 0.0089 | 0.0112 | 0.0138 | 0.0181 |
| BayesPCN | 0.0011 | 0.0033 | 0.0064 | 0.6606 |
| BayesPCN ($forget$) | 0.0026 | 0.0059 | 0.0108 | 0.0176 |

| Dropout CIFAR10 MSE | | | | |
|---|---|---|---|---|
| Sequence Length | 128 | 256 | 512 | 1024 |
| Identity | 1.1140 | 1.1178 | 1.1353 | 1.1481 |
| MHN | 0.0000 | 0.0000 | 0.0000 | 0.0000 |
| GPCN (Offline) | 0.0000 | 0.0000 | 0.0000 | 0.0001 |
| GPCN (Online) | 0.0022 | 0.0032 | 0.0053 | 0.0073 |
| BayesPCN | 0.0000 | 0.0000 | 0.0000 | 0.0001 |
| BayesPCN ($forget$) | 0.0000 | 0.0001 | 0.0005 | 0.0019 |

| Dropout Tiny ImageNet MSE | | | | |
|---|---|---|---|---|
| Sequence Length | 128 | 256 | 512 | 1024 |
| Identity | 1.0629 | 1.0889 | 1.1154 | 1.1072 |
| MHN | 0.0000 | 0.0000 | 0.0000 | 0.0000 |
| GPCN (Offline) | 0.0000 | 0.0000 | 0.0000 | 0.0000 |
| GPCN (Online) | 0.0036 | 0.0053 | 0.0069 | 0.0099 |
| BayesPCN | 0.0000 | 0.0000 | 0.0000 | 0.0000 |
| BayesPCN ($forget$) | 0.0000 | 0.0000 | 0.0002 | 0.0008 |

| Mask CIFAR10 MSE | | | | |
|---|---|---|---|---|
| Sequence Length | 128 | 256 | 512 | 1024 |
| Identity | 1.1272 | 1.1373 | 1.1619 | 1.1653 |
| MHN | 0.0000 | 0.0000 | 0.0011 | 0.0000 |
| GPCN (Offline) | 0.0000 | 0.0000 | 0.0000 | 0.0009 |
| GPCN (Online) | 0.0127 | 0.0255 | 0.0522 | 0.0791 |
| BayesPCN | 0.0000 | 0.0000 | 0.0001 | 0.0019 |
| BayesPCN ($forget$) | 0.0000 | 0.0008 | 0.0096 | 0.0465 |

| Mask Tiny ImageNet MSE | | | | |
|---|---|---|---|---|
| Sequence Length | 128 | 256 | 512 | 1024 |
| Identity | 1.0876 | 1.0884 | 1.0982 | 1.1132 |
| MHN | 0.0000 | 0.0000 | 0.0000 | 0.0010 |
| GPCN (Offline) | 0.0000 | 0.0000 | 0.0000 | 0.0001 |
| GPCN (Online) | 0.0081 | 0.0316 | 0.0441 | 0.0698 |
| BayesPCN | 0.0000 | 0.0000 | 0.0000 | 0.0000 |
| BayesPCN ($forget$) | 0.0000 | 0.0003 | 0.0031 | 0.0235 |

Table 1: Average MSE between the training images and the associative memory $read$ outputs for the white noise, dropout, and mask tasks on CIFAR10 and Tiny ImageNet. The reported values are from the best models of each category. Table 2 shows the same result with standard deviations.

evaluated 4-layer BayesPCN models, which we denote as BayesPCN ($forget$), that applied $forget$ with strength $\beta = 0.001$ to their weights once every 64 $write$ operations.

Table 1 shows the average MSE between the ground truth images and the associative memory $read$ outputs for models trained on data sequences of different lengths, where the reported scores come from the best models of each category. The results were averaged across three random seeds. Appendix I details how different hyperparameter values affect BayesPCN's $read$ performance.

BayesPCN's recall errors were on par with offline GPCN's recall errors and were orders of magnitude lower than online GPCN's recall errors on all hetero-associative recall tasks and sequence lengths. BayesPCN ($forget$) performed worse than BayesPCN in these settings but still outperformed online GPCN on all hetero-associative recall tasks and sequence lengths. On auto-associative recall tasks with sequences of length $\leq 512$, BayesPCN's recall errors were marginally higher than offline GPCN's recall errors but lower than online GPCN's recall errors. However, when the sequence length approached 1024, BayesPCN performed worse than online GPCN and on the TinyImageNet auto-associative recall task became even worse than the identity model. We hypothesize that BayesPCN's sudden performance drop stems from the fact that it is not "removing" old information unlike online GPCN and BayesPCN ($forget$) due to the conjugate Bayesian update's observation order invariance property. This causes BayesPCN to approach the overloaded memory regime, where recall of all data suffers, once enough information has been stored. Because BayesPCN ($forget$) deliberately removes old information and thus is farther away from the overloaded memory regime, it was able to outperform online GPCN on all auto-associative recall tasks and sequence lengths.

GPCN and BayesPCN models overall found hetero-associative recall easier than auto-associative recall, likely because the former fixes the key bits during memory $read$ and thus discourages the algorithm from wandering to less realistic $x^0$ values. Counterintuitively, GPCN and BayesPCN models were better at Tiny ImageNet recall over CIFAR10 recall. We attribute this behaviour to the fact that for some models, high-dimensional data recall can ironically be easier than low-dimensional data recall because individual datapoints are further apart from one another in higher dimensions.

A surprising discovery from this experiment was how good MHN's recall can be compared to GPCN and BayesPCN's recall when the memory query had a moderate amount of noise, a result inconsistent with the findings from Salvatori et al. [2021]. Because MHN explicitly retains past observations in a data matrix and its softmax inverse temperature was set very high ($\beta = 10,000$), MHN's recall

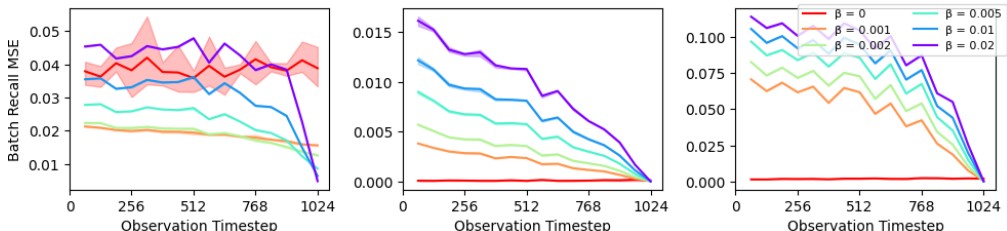

Figure 3: From left to right, a sample BayesPCN model's recall MSE for white noise ($\sigma = 0.2$), pixel dropout (25%), and pixel masking (25%) tasks with different forget strength $\beta$ after sequentially observing 1024 CIFAR10 images. The results were computed from three seeds and the shaded regions' upper and lower borders indicate the maximum and minimum errors observed from the runs.

effectively selected the observation that had the highest dot-product similarity to the memory query in our experiments. In Appendix G, we assess how MHN's recall compares to BayesPCN's recall when the memory query corruption is much higher. As expected, because the observation with the highest dot-product similarity to the memory query is often not the right image to recall, BayesPCN significantly outperformed MHN in hetero-associative recall tasks.

## 6.2 Forgetting

This section investigates the $forget$ operation's effect on BayesPCN's recall behaviour. Specifically, we show that $forget$ causes BayesPCN to better recall recent observations at the expense of older observations and that this effect is more pronounced the higher the forget strength is. We trained six structurally identical BayesPCN models on 1024 CIFAR10 images sequentially and applied $forget$ with strength $\beta = 0, 0.001, 0.002, 0.005, 0.01, 0.02$ to each model once after every 64 $write$ operations.

Figure 3 compares the models' recall MSE on 16 data batches of size 64 that were observed during different points of the training once the models have observed all 1024 datapoints. We see that BayesPCN with no $forget$ ($\beta = 0$) performs similarly on all datapoints regardless of their observation timestep, which is likely the consequence of conjugate Bayesian update's observation order invariance. On the other hand, BayesPCNs with $forget$ ($\beta > 0$) consistently perform better on newer datapoints with this effect being strongly correlated with the value of $\beta$. As previously shown, some amount of forgetting during training can improve the *overall* performance of the memory across all datapoints over no forgetting for long training sequences. Lastly, the higher the forget strength was the lower the recall error on the most recent data batch was for all tasks and forget strength values.

## 7 Discussion

One can intuitively think of BayesPCN $read$ as an approximate nearest neighbour operation with a learned similarity metric and $write$ as the continual learning of that metric. MHN on the other hand uses the dot-product distance as its similarity metric [Millidge et al., 2022]. This explains why MHN is not good at highly masked image recall because from a pixelwise dot-product distance perspective, highly masked image queries are often closer to unrelated images than to the ground truth images.

BayesPCN is closely related to the predictive coding theories of the brain's memory system. For example, Barron et al. [2020] suggests that the hippocampus may correspond to the top layer of a predictive coding network and the neo-cortical areas may correspond to the rest of the predictive coding network. Similarly, Dovrolis [2018] posits that their predictive coding network could serve as a model of the cortical column albeit without experimental results. It is however unclear how exactly BayesPCN's activation gradient descent and locally conjugate Bayesian weight update can be realized in the brain. Lastly, there are similarities between BayesPCN's diffusion-based $forget$ and the synaptic homeostasis hypothesis [Cirelli and Tononi, 2022], where during sleep synaptic weight strengths are decreased on average to facilitate future learning.

Aside from these biological connections, BayesPCN has many potential machine learning applications. One possible line of work is using it in conjunction with a neural controller that learns what

information to store into memory [Graves et al., 2014, Schlag et al., 2021]. Using BayesPCN in this way would confer the benefit of high query noise tolerance for neural controllers. Another line of work is using it to solve small data supervised learning problems, leveraging the fact that nearest neighbour methods work well when data availability is sparse [Ramsauer et al., 2020].

To conclude, we introduced BayesPCN, an associative memory model that can robustly retrieve past observations given various corrupted inputs, perform one-shot parameter update of a deep neural network to store new observations as attractors, and forget old observations to better store newer ones. We demonstrated that BayesPCN's recall performance is in most cases not significantly worse than the recall performance of its offline-learned counterpart even though it is continually learned. There are several intriguing extensions to BayesPCN that may address some of its limitations. Currently, BayesPCN $read$ can take up to thousands of activation gradient descent iterations to obtain the best recall result when a lot of data has been stored into memory. Supplementing BayesPCN $read$ with amortized inference may greatly reduce this computational requirement [Tschantz et al., 2022]. In addition, BayesPCN $read$ can return a nonsensical output if the memory query is too far away from the stored datapoints or if the memory is overloaded. Investigating BayesPCN's latent landscape may allow us design models with more robust basin of attractions that can easily be reached via activation gradient descent. Last but not least, extending BayesPCN beyond memorizing i.i.d. data to memorizing sequences like the human episodic memory would be of great interest.

## Acknowledgments and Disclosure of Funding

We would like to thank Boyan Beronov for the helpful discussions. In addition, we acknowledge the support of the Natural Sciences and Engineering Research Council of Canada (NSERC), the Canada CIFAR AI Chairs Program, and the Intel Parallel Computing Centers program. Additional support was provided by UBC's Composites Research Network (CRN), and Data Science Institute (DSI). This research was enabled in part by technical support and computational resources provided by WestGrid (www.westgrid.ca), Compute Canada (www.computecanada.ca), and Advanced Research Computing at the University of British Columbia (arc.ubc.ca).

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
