# A BayesPCN's Auto-Associative and Hetero-Associative *read*

---
**Algorithm 3** Auto-Associative Memory Read
---
1: **Input**: Memory log density $\log p(x^0, \mathbf{h}|x^0_{1:T})$, query vector $\bar{x}^0$
2: **Output**: Recall vector $x^{0*}$

---
3: Set the initial recall vector: $x^{0*} = \bar{x}^0$
4: **repeat**
5:    Find the most probable latent code given the input: $\mathbf{h}^* = \text{argmax}_\mathbf{h} \log p(x^{0*}, \mathbf{h}|x^0_{1:T})$
6:    Find the most probable input given the latent code: $x^{0*} = \text{argmax}_{x^0} \log p(x^0, \mathbf{h}^*|x^0_{1:T})$
7: **until** convergence or upper iteration limit

---

---
**Algorithm 4** Hetero-Associative Memory Read
---
1: **Input**: Memory log density $\log p(x^0, \mathbf{h}|x^0_{1:T})$, query vector $\bar{x}^0 = (k, v)$ where the key $k$ is known but the value $v$ is arbitrarily initialized
2: **Output**: Recall vector $x^{0*}$

---
3: Find the most probable value and latent code given the key:
4:    $v^*, \mathbf{h}^* = \text{argmax}_{v,\mathbf{h}} \log p((k, v), \mathbf{h}|x^0_{1:T})$
5: Set the final recall vector: $x^{0*} = (k, v^*)$

---

# B Derivation of BayesPCN *write*

$$p(\mathbf{W}|x^0_{1:t}) = \frac{1}{Z} p(\mathbf{W}|x^0_{1:t-1}) p(x^0_t|\mathbf{W}) \tag{12}$$

$$= \frac{1}{Z} p(\mathbf{W}|x^0_{1:t-1}) \mathbb{E}_{q(\mathbf{h}_t)}\left[\frac{p(x^0_t, \mathbf{h}_t|\mathbf{W})}{q(\mathbf{h}_t)}\right] \tag{13}$$

$$\approx \frac{1}{Z}\left[\sum_{n=1}^{N} \omega^{(n)}_{t-1} p^{(n)}(\mathbf{W}|x^0_{1:t-1}, \mathbf{h}^{(n)}_{1:t-1})\right] \mathbb{E}_{q(\mathbf{h}_t)}\left[\frac{p(x^0_t, \mathbf{h}_t|\mathbf{W})}{q(\mathbf{h}_t)}\right] \tag{14}$$

$$= \frac{1}{Z} \sum_{n=1}^{N} \omega^{(n)}_{t-1} p^{(n)}(\mathbf{W}|x^0_{1:t-1}, \mathbf{h}^{(n)}_{1:t-1}) \mathbb{E}_{q^{(n)}(\mathbf{h}_t)}\left[\frac{p(x^0_t, \mathbf{h}_t|\mathbf{W})}{q^{(n)}(\mathbf{h}_t)}\right] \tag{15}$$

$$\approx \frac{1}{Z} \sum_{n=1}^{N} \omega^{(n)}_{t-1} \frac{1}{S} \sum_{i=1}^{S} \frac{p^{(n)}(\mathbf{W}|x^0_{1:t-1}, \mathbf{h}^{(n)}_{1:t-1}) p(x^0_t, \mathbf{h}^{(n,i)}_t|\mathbf{W})}{q^{(n)}(\mathbf{h}^{(n,i)}_t)}, \quad \mathbf{h}^{(n,i)}_t \sim q^{(n)}(\mathbf{h}_t) \tag{16}$$

$$= \frac{1}{Z} \frac{1}{S} \sum_{n=1}^{N} \sum_{i=1}^{S} \omega^{(n)}_{t-1} \frac{p^{(n)}(x^0_t, \mathbf{h}^{(n,i)}_t|x^0_{1:t-1})}{q^{(n)}(\mathbf{h}^{(n,i)}_t)} p^{(n)}(\mathbf{W}|x^0_{1:t}, \mathbf{h}^{(n,i)}_{1:t}) \tag{17}$$

$$= \sum_{n=1}^{N} \sum_{i=1}^{S} \frac{\omega^{(n)}_{t-1} \frac{p^{(n)}(x^0_t, \mathbf{h}^{(n,i)}_t|x^0_{1:t-1})}{q^{(n)}(\mathbf{h}^{(n,i)}_t)}}{\sum_{n'=1}^{N} \sum_{i'=1}^{S} \omega^{(n')}_{t-1} \frac{p^{(n')}(x^0_t, \mathbf{h}^{(n',i')}_t|x^0_{1:t-1})}{q^{(n')}(\mathbf{h}^{(n',i')}_t)}} p^{(n)}(\mathbf{W}|x^0_{1:t}, \mathbf{h}^{(n,i)}_{1:t}) \tag{18}$$

$$= \sum_{n=1}^{N} \sum_{i=1}^{S} \omega^{(n,i)}_t p^{(n)}(\mathbf{W}|x^0_{1:t}, \mathbf{h}^{(n,i)}_{1:t}) \tag{19}$$

Equation 17 holds because

$$p^{(n)}(\mathbf{W}|x^0_{1:t}, \mathbf{h}^{(n)}_{1:t}) = \frac{p^{(n)}(\mathbf{W}|x^0_{1:t-1}, \mathbf{h}^{(n)}_{1:t-1}) p(x^0_t, \mathbf{h}^{(n,i)}_t|\mathbf{W})}{p^{(n)}(x^0_t, \mathbf{h}^{(n,i)}_t|x^0_{1:t-1})} \tag{20}$$

and Equation 18 holds because

$$Z = \int p(\mathbf{W}|x_{1:t-1}^0) \mathbb{E}_{q^{(n)}(\mathbf{h}_t)} \left[ \frac{p(x_t^0, \mathbf{h}_t|\mathbf{W})}{q^{(n)}(\mathbf{h}_t)} \right] d\mathbf{W} \tag{21}$$

$$\approx \int \frac{1}{S} \sum_{n=1}^{N} \sum_{i=1}^{S} \omega_{t-1}^{(n)} \frac{p^{(n)}(x_t^0, \mathbf{h}_t^{(n,i)}|x_{1:t-1}^0)}{q^{(n)}(\mathbf{h}_t^{(n,i)})} p^{(n)}(\mathbf{W}|x_{1:t}^0, \mathbf{h}_{1:t}^{(n,i)}) d\mathbf{W} \tag{22}$$

$$= \frac{1}{S} \sum_{n=1}^{N} \sum_{i=1}^{S} \omega_{t-1}^{(n)} \frac{p^{(n)}(x_t^0, \mathbf{h}_t^{(n,i)}|x_{1:t-1}^0)}{q^{(n)}(\mathbf{h}_t^{(n,i)})} \tag{23}$$

Finally, if $S = 1$ we get

$$\hat{p}(\mathbf{W}|x_{1:t}^0) = \sum_{n=1}^{N} \frac{\omega_{t-1}^{(n)} \frac{p^{(n)}(x_t^0, \mathbf{h}_t^{(n)}|x_{1:t-1}^0)}{q^{(n)}(\mathbf{h}_t^{(n)})}}{\sum_{n'=1}^{N} \omega_{t-1}^{(n')} \frac{p^{(n')}(x_t^0, \mathbf{h}_t^{(n')}|x_{1:t-1}^0)}{q^{(n')}(\mathbf{h}_t^{(n')})}} p^{(n)}(\mathbf{W}|x_{1:t}^0, \mathbf{h}_{1:t}^{(n)}) \tag{24}$$

∎

## C  Analytical Posterior Formulae for *write*

Let $z = f(x)$. Then, BayesPCN's top layer parameter update given $x^{0:L}$ is

$$\mu^* \leftarrow (\Sigma^{-1} + \frac{1}{\sigma_x^2}I)^{-1}(\Sigma^{-1}\mu + \frac{1}{\sigma_x^2}x^L) \tag{25}$$

$$\Sigma^* \leftarrow (\Sigma^{-1} + \frac{1}{\sigma_x^2}I)^{-1} \tag{26}$$

while the update for all other layers given $x^{0:L}$ is

$$R^{l*} \leftarrow R^l + U^l z^{l+1\top}(z^{l+1}U^l z^{l+1\top} + \sigma_x^2 I)^{-1}(x^l - z^{l+1}R^l) \tag{27}$$

$$U^{l*} \leftarrow U^l - U^l z^{l+1\top}(z^{l+1}U^l z^{l+1\top} + \sigma_x^2 I)^{-1}z^{l+1}U^l \tag{28}$$

## D  Analytical Diffusion Formulae for *forget*

Let $R_0^{1:L-1}, U_0^{1:L-1}, \mu_0, \Sigma_0$ be the memory prior parameters and $\beta$ be the forget strength. Then, BayesPCN's top layer diffusion update is

$$\mu^* = \sqrt{1-\beta}\mu + (1 - \sqrt{1-\beta})\mu_0 \tag{29}$$
$$\Sigma^* = (1-\beta)\Sigma + \beta\Sigma_0 \tag{30}$$

while the update for all other layers is

$$R^{l*} = \sqrt{1-\beta}R^l + (1 - \sqrt{1-\beta})R_0^l \tag{31}$$
$$U^{l*} = (1-\beta)U^l + \beta U_0^l \tag{32}$$

## E  Connections to Hopfield Networks

We show that modern Hopfield network's recall is equivalent to the recall of a BayesPCN model with $L = 0$. BayesPCN's activation log density is defined as $\log p(x^0, \mathbf{h}) = \log \left( \sum_{n=1}^{N} \omega^{(n)} p^{(n)}(x^0, \mathbf{h}) \right)$, and its gradient w.r.t. the activations is

$$\nabla_{x^0, \mathbf{h}} \log p(x^0, \mathbf{h}) = \sum_{n=1}^{N} \frac{\exp(\log \omega^{(n)} p^{(n)}(x^0, \mathbf{h}))}{\sum_{n'=1}^{N} \exp(\log \omega^{(n')} p^{(n')}(x^0, \mathbf{h}))} \nabla_{x^0, \mathbf{h}} \log p^{(n)}(x^0, \mathbf{h}), \tag{33}$$

On the other hand, MHN's energy is

$$E(q) = \frac{\beta}{2} q q^T - \log \sum_{j=1}^{N} \exp(\beta q K_j^T) \tag{34}$$

where $q \in \mathbb{R}^{1 \times d_k}$ is the query row vector, $K \in \mathbb{R}^{N \times d_k}$ is the key matrix, and $K_j \in \mathbb{R}^{1 \times d_k}$ is the j-th key row vector. $q$ and $K$ correspond to $x^0$ and $W^0$ in our paper's notation. The negative of the energy can be converted to the following Gaussian mixture log density.

$$\log p(q) = \log \left( \sum_{j=1}^{N} \frac{e^{\frac{\beta}{2} K_j K_j^T}}{\sum_{j'=1}^{N} e^{\frac{\beta}{2} K_{j'} K_{j'}^T}} \frac{1}{(2\pi\beta^{-1})^{\frac{d_k}{2}}} e^{-\frac{\beta}{2}(q-K_j)(q-K_j)^T} \right) \tag{35}$$

Recall in both BayesPCN and MHN is gradient ascent on the above log density. When we take the gradient with respect to the input vector $q$, we recover Equation 33.

$$\nabla_q \log p(q) = \sum_{j=1}^{N} \frac{e^{\frac{\beta}{2} q K_j^T}}{\sum_{j'=1}^{N} e^{\frac{\beta}{2} q K_{j'}^T}} \beta(K_j - q) \tag{36}$$

$$= \sum_{n=1}^{N} \frac{\exp(\log \omega^{(n)} p^{(n)}(x^0, \mathbf{h}))}{\sum_{n'=1}^{N} \exp(\log \omega^{(n')} p^{(n')}(x^0, \mathbf{h}))} \nabla_{x^0, \mathbf{h}} \log p^{(n)}(x^0, \mathbf{h}) \tag{37}$$

$$\text{where} \quad \omega^{(n)} = \frac{\beta}{2} K_n K_n^T, \quad \log p^{(n)}(x^0, \mathbf{h}) = \log \mathcal{N}(q; K_n, \frac{1}{\beta} I) \tag{38}$$

∎

We conclude that recall in Modern Hopfield Network is equivalent to recall under our framework, which is gradient descent on the log joint of a normal mixture w.r.t. neuron activations, where there are no hidden layers ($L = 0$). However training our model with $L = 0$ does not lead to the same memory update as the suggested training procedure of MHN, which is setting each key vector $K_j$ to some observed datapoint.

Universal Hopfield network [Millidge et al., 2022] proposes a framework for single-shot associative memory that decomposes recall into three components: similarity function, separation function, and projection matrix. Let $\mathbf{x} = (\bar{x}^0, \mathbf{h})$ be the row vector of all initial network activations when give the query $\bar{x}^0$. Equation 33 suggests that BayesPCN $read$'s implementation of those components is per particle weighted log joint $\log \omega^{(n)} p^{(n)}(\mathbf{x})$ of the query $\bar{x}^0$ for the similarity function, $softmax$ for

the separation function, and the matrix $\begin{bmatrix} \mathbf{x} + \gamma \nabla_{\mathbf{x}} \log p^{(1)}(\mathbf{x}) \\ \vdots \\ \mathbf{x} + \gamma \nabla_{\mathbf{x}} \log p^{(N)}(\mathbf{x}) \end{bmatrix}$ for the projection matrix where $\gamma$

is the learning rate. We can accommodate the fact that BayesPCN's $read$ does iterated conditional modes by zeroing out the fixed variable gradients in the projection matrix.

## F  Additional Experiment Details

All GPCN and BayesPCN models had $\sigma_W = 1, \sigma_x = 0.01$, and used Adam with learning rate 0.01 as the neuron activation gradient descent optimizer. All energy minimization w.r.t. the neuron activations took 500 gradient steps during the $write$ phase. During the $read$ phase, the outer loop iterated conditional mode in Algorithm 3 was repeated 30 times and the energy minimization in Algorithm 4 took $30 \times 500$ gradient steps. MHN models had $\beta = 10,000$ (equivalent to $\sigma_x = 0.01$), used Adam with learning rate 1.0 as the gradient descent optimizer, and performed gradient-descent based recall similar to BayesPCN based on the connection from Appendix E. All experiments were run on CIFAR10 and/or Tiny ImageNet datasets (both of which have the MIT License) and the image pixel values were normalized to fall between $[-1, 1]$. Offline GPCNs received 4000 iterations of network weight gradient descent steps. Online GPCNs took a single gradient step w.r.t. the network weights after the hidden activations converged per observation, a treatment consistent with that of the fast weight memory in Schlag et al. [2021]. BayesPCN $forget$ models had four hidden layers of width 1024, a single particle, and GELU activations. All hyperparameter ranges were chosen based on Salvatori et al. [2021]'s experiments and GPCN/BayesPCN's empirical results.

All experiments used NVIDIA Tesla V100 GPUs and were run on the university's internal clusters. Training the most expensive GPCN model (hidden layer width of 1024) and BayesPCN model (hidden layer width of 1024, 4 particles) on 1024 observations took 20 hours and 3 hours respectively. Evaluating the most expensive GPCN and BayesPCN models on all tasks took 25 minutes and 3 hours respectively.

| White Noise CIFAR10 MSE | | | | |
|---|---|---|---|---|
| Sequence Length | 128 | 256 | 512 | 1024 |
| Identity | $0.1596 \pm 0.0003$ | $0.1600 \pm 0.0001$ | $0.1600 \pm 0.0000$ | $0.1600 \pm 0.0001$ |
| MHN | $0.0000 \pm 0.0000$ | $0.0000 \pm 0.0000$ | $0.0000 \pm 0.0000$ | $0.0000 \pm 0.0000$ |
| GPCN (Offline) | $0.0028 \pm 0.0000$ | $0.0046 \pm 0.0001$ | $0.0073 \pm 0.0000$ | $0.0121 \pm 0.0001$ |
| GPCN (Online) | $0.0103 \pm 0.0001$ | $0.0150 \pm 0.0001$ | $0.0191 \pm 0.0000$ | $0.0210 \pm 0.0001$ |
| BayesPCN | $0.0017 \pm 0.0003$ | $0.0085 \pm 0.0002$ | $0.0146 \pm 0.0001$ | $0.0337 \pm 0.0007$ |
| BayesPCN ($forget$) | $0.0064 \pm 0.0001$ | $0.0102 \pm 0.0001$ | $0.0145 \pm 0.0001$ | $0.0188 \pm 0.0002$ |

| White Noise Tiny ImageNet MSE | | | | |
|---|---|---|---|---|
| Sequence Length | 128 | 256 | 512 | 1024 |
| Identity | $0.1600 \pm 0.0000$ | $0.1602 \pm 0.0000$ | $0.1601 \pm 0.0000$ | $0.1600 \pm 0.0001$ |
| MHN | $0.0000 \pm 0.0000$ | $0.0000 \pm 0.0000$ | $0.0000 \pm 0.0000$ | $0.0000 \pm 0.0000$ |
| GPCN (Offline) | $0.0005 \pm 0.0000$ | $0.0010 \pm 0.0000$ | $0.0018 \pm 0.0000$ | $0.0067 \pm 0.0004$ |
| GPCN (Online) | $0.0089 \pm 0.0002$ | $0.0112 \pm 0.0002$ | $0.0138 \pm 0.0001$ | $0.0181 \pm 0.0001$ |
| BayesPCN | $0.0011 \pm 0.0001$ | $0.0033 \pm 0.0000$ | $0.0064 \pm 0.0001$ | $0.6606 \pm 0.0267$ |
| BayesPCN ($forget$) | $0.0026 \pm 0.0000$ | $0.0059 \pm 0.0000$ | $0.0108 \pm 0.0001$ | $0.0176 \pm 0.0001$ |

| Dropout CIFAR10 MSE | | | | |
|---|---|---|---|---|
| Sequence Length | 128 | 256 | 512 | 1024 |
| Identity | $1.1140 \pm 0.0059$ | $1.1178 \pm 0.0009$ | $1.1353 \pm 0.0014$ | $1.1481 \pm 0.0010$ |
| MHN | $0.0000 \pm 0.0000$ | $0.0000 \pm 0.0000$ | $0.0000 \pm 0.0000$ | $0.0000 \pm 0.0000$ |
| GPCN (Offline) | $0.0000 \pm 0.0000$ | $0.0000 \pm 0.0000$ | $0.0000 \pm 0.0000$ | $0.0001 \pm 0.0000$ |
| GPCN (Online) | $0.0022 \pm 0.0000$ | $0.0032 \pm 0.0001$ | $0.0053 \pm 0.0001$ | $0.0073 \pm 0.0000$ |
| BayesPCN | $0.0000 \pm 0.0000$ | $0.0000 \pm 0.0000$ | $0.0000 \pm 0.0000$ | $0.0001 \pm 0.0000$ |
| BayesPCN ($forget$) | $0.0000 \pm 0.0000$ | $0.0001 \pm 0.0000$ | $0.0005 \pm 0.0000$ | $0.0019 \pm 0.0000$ |

| Dropout Tiny ImageNet MSE | | | | |
|---|---|---|---|---|
| Sequence Length | 128 | 256 | 512 | 1024 |
| Identity | $1.0629 \pm 0.0014$ | $1.0889 \pm 0.0006$ | $1.1154 \pm 0.0005$ | $1.1072 \pm 0.0008$ |
| MHN | $0.0000 \pm 0.0000$ | $0.0000 \pm 0.0000$ | $0.0000 \pm 0.0000$ | $0.0000 \pm 0.0000$ |
| GPCN (Offline) | $0.0000 \pm 0.0000$ | $0.0000 \pm 0.0000$ | $0.0000 \pm 0.0000$ | $0.0000 \pm 0.0000$ |
| GPCN (Online) | $0.0036 \pm 0.0002$ | $0.0053 \pm 0.0002$ | $0.0069 \pm 0.0001$ | $0.0099 \pm 0.0001$ |
| BayesPCN | $0.0000 \pm 0.0000$ | $0.0000 \pm 0.0000$ | $0.0000 \pm 0.0000$ | $0.0000 \pm 0.0000$ |
| BayesPCN ($forget$) | $0.0000 \pm 0.0000$ | $0.0000 \pm 0.0000$ | $0.0002 \pm 0.0000$ | $0.0008 \pm 0.0000$ |

| Mask CIFAR10 MSE | | | | |
|---|---|---|---|---|
| Sequence Length | 128 | 256 | 512 | 1024 |
| Identity | $1.1272 \pm 0.0000$ | $1.1373 \pm 0.0000$ | $1.1619 \pm 0.0000$ | $1.1653 \pm 0.0000$ |
| MHN | $0.0000 \pm 0.0000$ | $0.0000 \pm 0.0000$ | $0.0011 \pm 0.0000$ | $0.0000 \pm 0.0000$ |
| GPCN (Offline) | $0.0000 \pm 0.0000$ | $0.0000 \pm 0.0000$ | $0.0000 \pm 0.0000$ | $0.0009 \pm 0.0000$ |
| GPCN (Online) | $0.0127 \pm 0.0008$ | $0.0255 \pm 0.0009$ | $0.0522 \pm 0.0006$ | $0.0791 \pm 0.0005$ |
| BayesPCN | $0.0000 \pm 0.0000$ | $0.0000 \pm 0.0000$ | $0.0001 \pm 0.0000$ | $0.0019 \pm 0.0000$ |
| BayesPCN ($forget$) | $0.0000 \pm 0.0000$ | $0.0008 \pm 0.0000$ | $0.0096 \pm 0.0001$ | $0.0465 \pm 0.0001$ |

| Mask Tiny ImageNet MSE | | | | |
|---|---|---|---|---|
| Sequence Length | 128 | 256 | 512 | 1024 |
| Identity | $1.0876 \pm 0.0000$ | $1.0884 \pm 0.0000$ | $1.0982 \pm 0.0000$ | $1.1132 \pm 0.0000$ |
| MHN | $0.0000 \pm 0.0000$ | $0.0000 \pm 0.0000$ | $0.0000 \pm 0.0000$ | $0.0010 \pm 0.0000$ |
| GPCN (Offline) | $0.0000 \pm 0.0000$ | $0.0000 \pm 0.0000$ | $0.0000 \pm 0.0000$ | $0.0001 \pm 0.0000$ |
| GPCN (Online) | $0.0081 \pm 0.0007$ | $0.0316 \pm 0.0033$ | $0.0441 \pm 0.0012$ | $0.0698 \pm 0.0010$ |
| BayesPCN | $0.0000 \pm 0.0000$ | $0.0000 \pm 0.0000$ | $0.0000 \pm 0.0000$ | $0.0000 \pm 0.0000$ |
| BayesPCN ($forget$) | $0.0000 \pm 0.0000$ | $0.0003 \pm 0.0000$ | $0.0031 \pm 0.0000$ | $0.0235 \pm 0.0001$ |

Table 2: Table 1 results with standard deviations calculated across three seeds.

# G BayesPCN vs MHN Recall on Highly Noised Queries

This section reports the recall performance of MHN and BayesPCN models on high query noise associative recall tasks. The experiment setup is similar to that of Section 6 aside from the fact that the white noise tasks had the noise standard deviation set to $0.8$ instead of $0.2$, the dropout tasks randomly blacked out $75\%$ of the pixels instead of $25\%$, and the masking tasks blacked out $75\%$ of the rightmost pixels instead of $25\%$. Because the high query noise task is harder, we show the recall result after the models observed 16, 32, 64, and 128 datapoints. BayesPCN models had four hidden layers of width 256, a single particle, and GELU activations. MHNs again used $\beta = 10,000$.

| White Noise CIFAR10 MSE | | | | |
|---|---|---|---|---|
| Sequence Length | 16 | 32 | 64 | 128 |
| Identity | $2.5564 \pm 0.0086$ | $2.5525 \pm 0.0023$ | $2.5586 \pm 0.0105$ | $2.5565 \pm 0.0052$ |
| MHN | $0.0086 \pm 0.0000$ | $0.0023 \pm 0.0000$ | $0.0105 \pm 0.0000$ | $0.0052 \pm 0.0000$ |
| BayesPCN | $0.0111 \pm 0.0003$ | $0.0203 \pm 0.0002$ | $0.0394 \pm 0.0007$ | $0.0755 \pm 0.0002$ |

| White Noise Tiny ImageNet MSE | | | | |
|---|---|---|---|---|
| Sequence Length | 16 | 32 | 64 | 128 |
| Identity | $2.5586 \pm 0.0105$ | $2.5565 \pm 0.0052$ | $2.5584 \pm 0.0047$ | $2.5582 \pm 0.0036$ |
| MHN | $0.0000 \pm 0.0000$ | $0.0000 \pm 0.0000$ | $0.0000 \pm 0.0000$ | $0.0000 \pm 0.0000$ |
| BayesPCN | $0.0033 \pm 0.0001$ | $0.0064 \pm 0.0002$ | $0.0125 \pm 0.0003$ | $0.0242 \pm 0.0002$ |

| Dropout CIFAR10 MSE | | | | |
|---|---|---|---|---|
| Sequence Length | 16 | 32 | 64 | 128 |
| Identity | $1.0060 \pm 0.0054$ | $1.0276 \pm 0.0044$ | $1.0691 \pm 0.0026$ | $1.1095 \pm 0.0018$ |
| MHN | $0.3517 \pm 0.0027$ | $0.3596 \pm 0.0029$ | $0.3635 \pm 0.0031$ | $0.3840 \pm 0.0010$ |
| BayesPCN | $0.0000 \pm 0.0000$ | $0.0000 \pm 0.0000$ | $0.0000 \pm 0.0000$ | $0.0000 \pm 0.0000$ |

| Dropout Tiny ImageNet MSE | | | | |
|---|---|---|---|---|
| Sequence Length | 16 | 32 | 64 | 128 |
| Identity | $1.0365 \pm 0.0022$ | $1.0746 \pm 0.0012$ | $1.1418 \pm 0.0007$ | $1.0625 \pm 0.0011$ |
| MHN | $0.4967 \pm 0.0026$ | $0.5096 \pm 0.0006$ | $0.5741 \pm 0.0021$ | $0.5630 \pm 0.0036$ |
| BayesPCN | $0.0000 \pm 0.0000$ | $0.0000 \pm 0.0000$ | $0.0000 \pm 0.0000$ | $0.0000 \pm 0.0000$ |

| Mask CIFAR10 MSE | | | | |
|---|---|---|---|---|
| Sequence Length | 16 | 32 | 64 | 128 |
| Identity | $0.9686 \pm 0.0000$ | $0.9941 \pm 0.0000$ | $1.0563 \pm 0.0000$ | $1.0987 \pm 0.0000$ |
| MHN | $0.3361 \pm 0.0000$ | $0.3315 \pm 0.0000$ | $0.3650 \pm 0.0000$ | $0.3957 \pm 0.0000$ |
| BayesPCN | $0.0000 \pm 0.0000$ | $0.0001 \pm 0.0000$ | $0.0001 \pm 0.0000$ | $0.0006 \pm 0.0000$ |

| Mask Tiny ImageNet MSE | | | | |
|---|---|---|---|---|
| Sequence Length | 16 | 32 | 64 | 128 |
| Identity | $1.0019 \pm 0.0000$ | $1.0606 \pm 0.0000$ | $1.1370 \pm 0.0000$ | $1.0597 \pm 0.0000$ |
| MHN | $0.4534 \pm 0.0000$ | $0.4896 \pm 0.0000$ | $0.7033 \pm 0.0000$ | $0.6378 \pm 0.0000$ |
| BayesPCN | $0.0000 \pm 0.0000$ | $0.0000 \pm 0.0000$ | $0.0000 \pm 0.0000$ | $0.0001 \pm 0.0000$ |

Table 3: Average MSE between the training images and the associative memory $read$ outputs for the high query noise white noise, dropout, and mask tasks on CIFAR10 and Tiny ImageNet datasets.

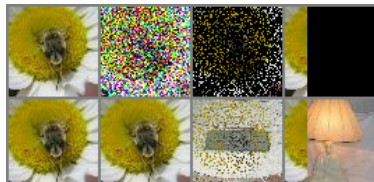 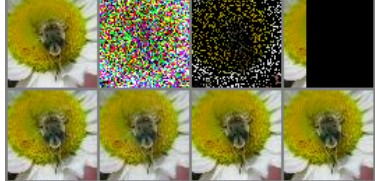

Figure 4: Sample MHN (**left**) and BayesPCN (**right**) $read$ outputs for the white noise ($\sigma = 0.8$), pixel dropout (75%), and pixel masking (75%) tasks. The top row contains the memory $read$ inputs and the bottom row contains the memory $read$ outputs.

# H   Additional Qualitative Results

Figure 5 qualitatively demonstrates how BayesPCN's read scales with the number of stored datapoints for the CIFAR10 recall tasks. BayesPCN models are able to output images that are very close to the original image even when the inputs are significantly corrupted. As the number of observations increases, the $read$ operation is still able to reconstruct the original image but gets worse at recovering the original image given a corrupted version of it.

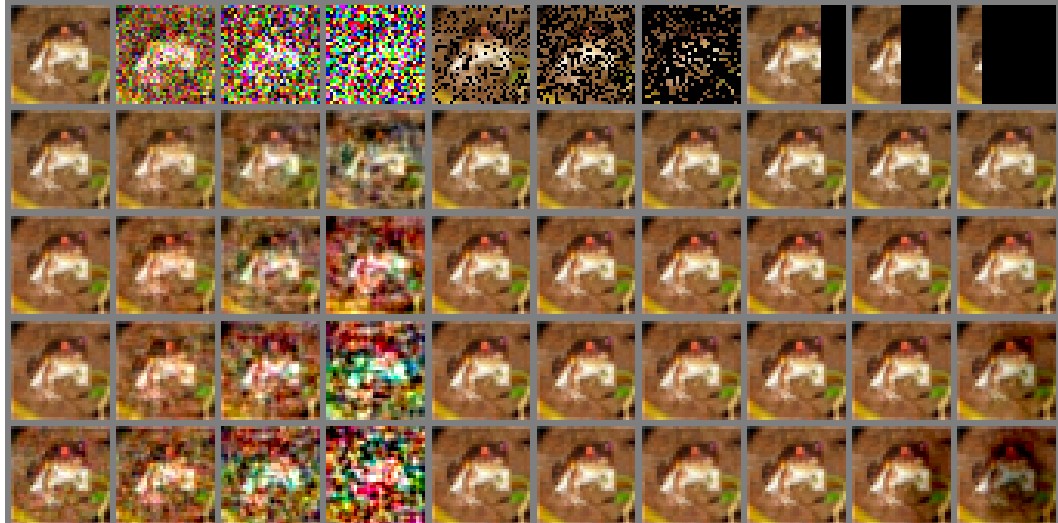

Figure 5: Example BayesPCN recall result from the CIFAR10 auto-associative and hetero-associative tasks. From top to bottom rows are the comparisons of BayesPCN's $read$ inputs and outputs where the task is to recover the first observation seen during the $write$ phase (the top left frog image) after sequentially observing 127, 255, 511, and 1023 additional datapoints.

# I   Effects of BayesPCN Hyperparameters

## I.1   Network Width and Depth

Figure 6 and Table 4 illustrate how BayesPCN's recall accuracy and MSE scale with the network width and depth. A recall is considered correct if the MSE between the ground truth and the recalled data is less than $0.01$. BayesPCN models had GELU activation functions, $\sigma_W = 1$, $\sigma_x = 0.01$, and a single particle.

We found that the increased network width was helpful across all tasks. Increased network depth was helpful when moving from network depth of 2 to 4, but moving from network depth of 4 to 8 had no noticeable impact across all tasks. We note that because the Figure 6 depicts recall accuracy not MSE, if the memory performance generally declines and the average recall MSE exceeds $0.01$, this can lead to very low accuracy even if the actual recall MSE is not much greater than $0.01$.

## I.2   Network Weight Prior Uncertainty and Observation Noise

We also investigate the effect of $\sigma_W$ and $\sigma_x$ hyperparameters on BayesPCN's scaling properties. Since $\sigma_W$ determines the prior uncertainty and $\sigma_x$ determines the observation noise, higher $\sigma_W$ and lower $\sigma_x$ reduces the prior's impact and increases the new observation's impact on the network weight's posterior. Hence, we can control the memory $write$ strength by modulating $\sigma_W, \sigma_x$.

Table 5 describes the CIFAR10 recall results of nine structurally identical BayesPCN models with four hidden layers of size 1024, a single particle, and GELU activations but with different values of $\sigma_W$ and $\sigma_x$. We observe that lower $\sigma_W$ and higher $\sigma_x$ tend to alleviate the memory overloading behaviour. We hypothesize that this is the case because lower $\sigma_W$ and higher $\sigma_x$ encourage the synaptic weights' Frobenius norms to remain small, causing activation gradient descent more stable.

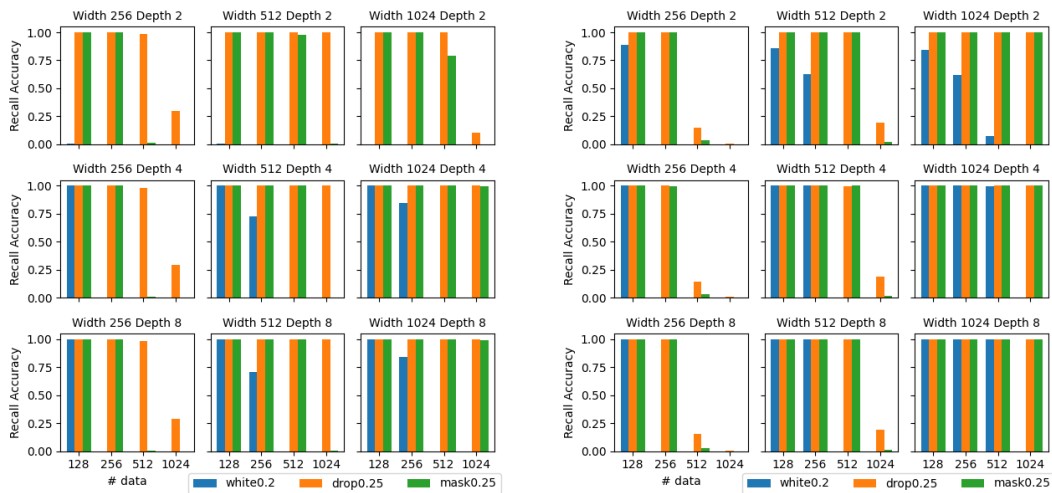

Figure 6: Recall accuracy of BayesPCN with different network width and depth on CIFAR10 (**left**) and Tiny ImageNet (**right**) tasks.

| White Noise CIFAR10 MSE | | | | |
|---|---|---|---|---|
| Sequence Length | 128 | 256 | 512 | 1024 |
| BayesPCN L2 | $0.0284 \pm 0.0001$ | $0.0445 \pm 0.0000$ | $0.0470 \pm 0.0001$ | $0.0830 \pm 0.0034$ |
| BayesPCN L4 | $0.0058 \pm 0.0001$ | $0.0092 \pm 0.0001$ | $0.0146 \pm 0.0001$ | $0.0337 \pm 0.0007$ |
| BayesPCN L8 | $0.0058 \pm 0.0001$ | $0.0092 \pm 0.0001$ | $0.0146 \pm 0.0000$ | $0.0344 \pm 0.0015$ |

| White Noise Tiny ImageNet MSE | | | | |
|---|---|---|---|---|
| Sequence Length | 128 | 256 | 512 | 1024 |
| BayesPCN L2 | $0.0083 \pm 0.0001$ | $0.0096 \pm 0.0001$ | $0.0178 \pm 0.0001$ | $0.3458 \pm 0.1281$ |
| BayesPCN L4 | $0.0020 \pm 0.0001$ | $0.0037 \pm 0.0001$ | $0.0066 \pm 0.0002$ | $12.4499 \pm 1.1542$ |
| BayesPCN L8 | $0.0020 \pm 0.0001$ | $0.0036 \pm 0.0001$ | $0.0065 \pm 0.0002$ | $13.8584 \pm 1.3771$ |

| Dropout CIFAR10 MSE | | | | |
|---|---|---|---|---|
| Sequence Length | 128 | 256 | 512 | 1024 |
| BayesPCN L2 | $0.0000 \pm 0.0000$ | $0.0000 \pm 0.0000$ | $0.0000 \pm 0.0000$ | $0.0142 \pm 0.0000$ |
| BayesPCN L4 | $0.0000 \pm 0.0000$ | $0.0000 \pm 0.0000$ | $0.0000 \pm 0.0000$ | $0.0001 \pm 0.0000$ |
| BayesPCN L8 | $0.0000 \pm 0.0000$ | $0.0000 \pm 0.0000$ | $0.0000 \pm 0.0000$ | $0.0001 \pm 0.0000$ |

| Dropout Tiny ImageNet MSE | | | | |
|---|---|---|---|---|
| Sequence Length | 128 | 256 | 512 | 1024 |
| BayesPCN L2 | $0.0000 \pm 0.0000$ | $0.0000 \pm 0.0000$ | $0.0000 \pm 0.0000$ | $0.0000 \pm 0.0000$ |
| BayesPCN L4 | $0.0000 \pm 0.0000$ | $0.0000 \pm 0.0000$ | $0.0000 \pm 0.0000$ | $0.0000 \pm 0.0000$ |
| BayesPCN L8 | $0.0000 \pm 0.0000$ | $0.0000 \pm 0.0000$ | $0.0000 \pm 0.0000$ | $0.0006 \pm 0.0006$ |

| Mask CIFAR10 MSE | | | | |
|---|---|---|---|---|
| Sequence Length | 128 | 256 | 512 | 1024 |
| BayesPCN L2 | $0.0000 \pm 0.0000$ | $0.0002 \pm 0.0000$ | $0.0081 \pm 0.0027$ | $0.1024 \pm 0.0001$ |
| BayesPCN L4 | $0.0000 \pm 0.0000$ | $0.0000 \pm 0.0000$ | $0.0001 \pm 0.0000$ | $0.0019 \pm 0.0000$ |
| BayesPCN L8 | $0.0000 \pm 0.0000$ | $0.0000 \pm 0.0000$ | $0.0001 \pm 0.0000$ | $0.0019 \pm 0.0000$ |

| Mask Tiny ImageNet MSE | | | | |
|---|---|---|---|---|
| Sequence Length | 128 | 256 | 512 | 1024 |
| BayesPCN L2 | $0.0000 \pm 0.0000$ | $0.0000 \pm 0.0000$ | $0.0001 \pm 0.0000$ | $0.0004 \pm 0.0000$ |
| BayesPCN L4 | $0.0000 \pm 0.0000$ | $0.0000 \pm 0.0000$ | $0.0000 \pm 0.0000$ | $0.0000 \pm 0.0000$ |
| BayesPCN L8 | $0.0000 \pm 0.0000$ | $0.0000 \pm 0.0000$ | $0.0000 \pm 0.0000$ | $0.0000 \pm 0.0000$ |

Table 4: Average MSE between the training images and the associative memory $read$ outputs for the high query noise white noise, dropout, and mask tasks on CIFAR10 and Tiny ImageNet datasets.

| White Noise CIFAR10 MSE | | | | | | | | |
|---|---|---|---|---|---|---|---|---|
| $\sigma_W$ | $\sigma_x$ | 16 | 32 | 64 | 128 | 256 | 512 | 1024 |
| 0.5 | 0.05 | 0.0294 | 0.0304 | 0.0304 | 0.0289 | 0.0256 | 0.0222 | 0.0189 |
| 0.5 | 0.01 | 0.0012 | 0.0018 | 0.0031 | 0.0052 | 0.0083 | 0.0133 | 0.0213 |
| 0.5 | 0.005 | 0.0012 | 0.0018 | 0.0032 | 0.0055 | 0.0095 | 0.0169 | 1.859 |
| 1.0 | 0.05 | 0.0250 | 0.0264 | 0.0267 | 0.0256 | 0.0230 | 0.0204 | 0.0180 |
| 1.0 | 0.01 | 0.0014 | 0.0021 | 0.0036 | 0.0058 | 0.0091 | 0.0146 | 0.0329 |
| 1.0 | 0.005 | 0.0011 | 0.0045 | 0.0166 | 0.0325 | 0.0361 | 0.0328 | 2.8281 |
| 5.0 | 0.05 | 0.0258 | 0.0285 | 0.0279 | 0.0324 | 0.0814 | 0.1494 | 0.7720 |
| 5.0 | 0.01 | 0.2567 | 1.0674 | 1.5191 | 1.8586 | 5.7848 | 13.1495 | 29.4737 |
| 5.0 | 0.005 | 0.3738 | 1.4906 | 2.2916 | 3.0278 | 14.7747 | 99.3011 | 79.1281 |

| Dropout CIFAR10 MSE | | | | | | | | |
|---|---|---|---|---|---|---|---|---|
| $\sigma_W$ | $\sigma_x$ | 16 | 32 | 64 | 128 | 256 | 512 | 1024 |
| 0.5 | 0.05 | 0.0001 | 0.0002 | 0.0002 | 0.0004 | 0.0007 | 0.0015 | 0.0025 |
| 0.5 | 0.01 | 0.0000 | 0.0000 | 0.0000 | 0.0000 | 0.0000 | 0.0000 | 0.0001 |
| 0.5 | 0.005 | 0.0000 | 0.0000 | 0.0000 | 0.0000 | 0.0000 | 0.0000 | 0.0000 |
| 1.0 | 0.05 | 0.0001 | 0.0002 | 0.0002 | 0.0004 | 0.0007 | 0.0015 | 0.0025 |
| 1.0 | 0.01 | 0.0000 | 0.0000 | 0.0000 | 0.0000 | 0.0000 | 0.0000 | 0.0001 |
| 1.0 | 0.005 | 0.0000 | 0.0000 | 0.0000 | 0.0000 | 0.0000 | 0.0001 | 0.0262 |
| 5.0 | 0.05 | 0.0001 | 0.0001 | 0.0002 | 0.0003 | 0.0005 | 0.0009 | 0.0153 |
| 5.0 | 0.01 | 0.0001 | 0.0001 | 0.0001 | 0.0002 | 0.0311 | 0.1426 | 0.2543 |
| 5.0 | 0.005 | 0.0000 | 0.0001 | 0.0001 | 0.0002 | 0.0274 | 0.1984 | 0.7999 |

| Mask CIFAR10 MSE | | | | | | | | |
|---|---|---|---|---|---|---|---|---|
| $\sigma_W$ | $\sigma_x$ | 16 | 32 | 64 | 128 | 256 | 512 | 1024 |
| 0.5 | 0.05 | 0.0002 | 0.0004 | 0.0008 | 0.0017 | 0.0048 | 0.0114 | 0.0239 |
| 0.5 | 0.01 | 0.0000 | 0.0000 | 0.0000 | 0.0000 | 0.0000 | 0.0002 | 0.0028 |
| 0.5 | 0.005 | 0.0000 | 0.0000 | 0.0000 | 0.0000 | 0.0000 | 0.0000 | 0.0002 |
| 1.0 | 0.05 | 0.0002 | 0.0003 | 0.0007 | 0.0015 | 0.0046 | 0.0112 | 0.0240 |
| 1.0 | 0.01 | 0.0000 | 0.0000 | 0.0000 | 0.0000 | 0.0000 | 0.0001 | 0.0019 |
| 1.0 | 0.005 | 0.0000 | 0.0000 | 0.0000 | 0.0000 | 0.0000 | 0.0003 | 0.2248 |
| 5.0 | 0.05 | 0.0001 | 0.0003 | 0.0005 | 0.0010 | 0.0026 | 0.0061 | 0.2153 |
| 5.0 | 0.01 | 0.0000 | 0.0000 | 0.0001 | 0.0002 | 0.1472 | 0.7554 | 1.8079 |
| 5.0 | 0.005 | 0.0000 | 0.0000 | 0.0001 | 0.0001 | 0.1713 | 0.8259 | 6.8712 |

Table 5: Average MSE between the training images and the associative memory *read* outputs for BayesPCN models with different $\sigma_W, \sigma_x$ hyperparameters after observing 16, 32, 64, 128, 256, 512, and 1024 CIFAR10 images.

As an aside, the white noise recall MSE of BayesPCN with $\sigma_x = 0.05$ and $\sigma_W \in \{0.5, 1.0\}$ decreased as more datapoints were observed from sequence length 64 and onward. On visual inspection, we found that the model's auto-associative recall outputs for both observed and unobserved inputs became less blurry as more datapoints were written into memory. We hypothesize that the model learned to generalize at some point of its training.

# J    BayesPCN Generalization

Figure 7 illustrates BayesPCN's read outputs for unseen image queries after different number of datapoints have been stored into memory. As BayesPCN observes more data, it learns to "generalize" and gets better at reconstructing and even mildly removing white noise from unseen images. This can be attributed to the model continual learning its internal representation that better "describe" the data distribution. We expected this behaviour to occur since S-NCN [Ororbia et al., 2019], a model similar in structure to GPCN, could continually learn to perform discriminative tasks.

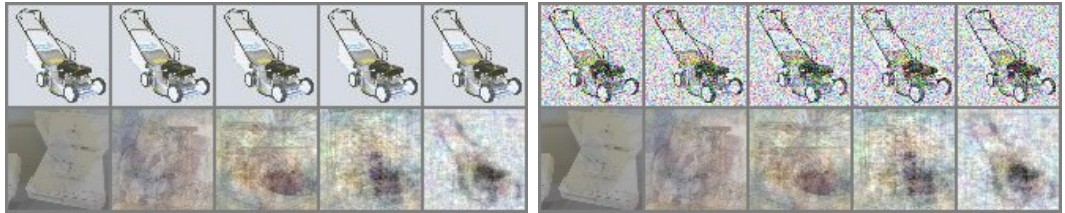

Figure 7: BayesPCN's $read$ outputs given ground truth (**left**) and noised (**right**) unseen images as inputs after observing $2, 8, 32, 128, 512$ training images.

## K   BayesPCN Sampling

Both GPCN and BayesPCN at the core are as much generative models as they are associative memories. We examine the quality of ancestral sampling samples from both models in Figure 8. When ancestral sampling, BayesPCN did not marginalize out the synaptic weights and instead fixed them to the mean parameters of $p(\mathbf{W}|x_{1:t-1}^0, \mathbf{h}_{1:t-1}^{(n)})$.

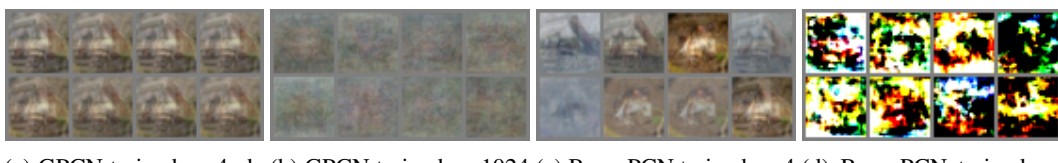

(a) GPCN trained on 4 observations.

(b) GPCN trained on 1024 observations.

(c) BayesPCN trained on 4 observations.

(d) BayesPCN trained on 1024 observations.

Figure 8: Ancestral sampling results from GPCN and BayesPCN models trained on CIFAR10.

We find that both GPCN and BayesPCN samples are superpositions of the training images. However, as BayesPCN is trained on more and more observations, its sample quality quickly deteriorates. We hypothesize that the poor sample quality for both GPCN and BayesPCN stems from the approximate nature of their parameter estimation. For example, BayesPCN's particle count would have to be much greater than 4 to accurately capture the true posterior $p(\mathbf{W}|x_{1:t-1}^0)$ using its sequential importance sampling estimate and the variational distribution over the hidden activations should not be Dirac distributed. However, we note that Ororbia and Kifer [2022] has successfully trained predictive coding networks to be good generative models.