# OpenReview forum: "BayesPCN: A Continually Learnable Predictive Coding Associative Memory"
_NeurIPS.cc/2022/Conference — NeurIPS 2022 Accept_

### Official Review · Reviewer_HskX · 2022-07-07

**Rating:** 5
**Confidence:** 5
**Soundness:** 2 fair
**Presentation:** 2 fair
**Contribution:** 2 fair

**Summary:**

This paper proposes an interesting extension of PC-based memory models, that tackles an existing problem of the original generative network: the inability to remember stored memories when presented with new ones, i.e., to write new memories without resetting the model. Generative predictive coding networks were, in fact, trained only once using full batch training on the whole dataset. This work tackles this problem by introducing a new method to write memories in the database. Additionally, it allows to use a similar method to implement a 'forget' operation, that cancels memories from the database.

**Questions:**

In Table 1, you show the average MSE of every data point, under specific kinds of corruption. However, what is the MSE of the first stored datapoints when all the patterns are stored? What is the MSE of the last ones?
Particularly, I have the feeling that the reported MSE only show the weakness of the model when confronted with the other baselines, and no experiment shows the advantages. You could perform the same experiments using the standard generative network trained via running inference, and make it memorize one datapoint at the time, exactly how you did for BayesPCNs. I would guess that your model would perform much better on the retrieval of past memories. Why is something like that not included?



Line 237: "A surprising discovery from this experiment was how good MHN’s recall is compared to GPCN and BayesPCN when the memory read input has a moderate amount of noise."
This follows as the induced Gaussian noise has mean zero, and hence the expected normalized dot product between a memory + Gaussian noise and the original memory is 1. As MHNs use dot products to perform similarity scores, they actually perform quite well under this very specific task. This, however, does not happen if the vector of the pixels is not normalised before performing the MHNs operation: in this case, the model will often return the 'whitest' image of the dataset, as the colour white is the one with the highest norm in the rgb collar system.


Fig.1 is slightly confusing: the description reads nicely and well explains the point, but the image is just a bunch of pictures. I would improve it by adding some pointers directly in the Figure, as well as removing the forth line.

**Ethics Review Area:**

["Legal Compliance (e.g., GDPR, copyright, terms of use)"]

**Limitations:**

Yes they have.

**Strengths And Weaknesses:**

Strengths:

1) The model works well, and is indeed able to store memories sequentially without the need of re-training the model from scratch, or without a loss in performance on memories stored multiple 'write' operations before;

2) This method could be potentially interesting in context that are more general than a generative model to perform associative memories, such as continual learning;

3) The literature and related work is well covered.

Weaknesses:

1) I personally found the description of the 'write' operation to be unclear, and it took me a couple of readings to fully understand it: The role of the n-th particle weight $\omega^{n}$ is not well explained, and the same applies to the n-th model $p^{n}$. Hence, a better and clearer explanation seems needed. It also says that these concepts are introduced in Sec.3, where exactly?

2) The experiments seem to not really show the potential and limitations of the model on the task that concerns this paper the most: continual 'writing' of memories. I will list below a couple of questions I have about the performance of the model that are both a personal curiosity, and a tip on some experiments that would make the paper more (in my opinion) interesting.

These are my main concern, and a rebuttal that addresses the above two points would make me increase the final score.

---

> ### Author Response · Authors · 2022-08-02
> **Response to Reviewer 4**
>
> We would like to thank the reviewer for their time and feedback. We address the points individually and kindly ask the reviewer to increase his/her score based on our response.
>
> - “Description of the ‘write’ operation is unclear, the role of particles are not well explained, clarify link to section 3”
>   - We have revised the memory write section and clarified its link to the model presented in section 3 to make it easier to understand.
> - “Can you show the MSE of the first and last datapoints”
>   - We will do this in the camera-ready version of the paper. We do note that without forgetting, the MSE of the first and last datapoints were not very different. Figure 2 showcases this when forgetting is applied.
> - “Experiments only show weakness vs the baselines and not advantages”
>   - To better highlight the advantages of BayesPCN we included Table 3 that explicitly shows BayesPCN outperforming MHN by a large margin on high query noise hetero-associative recall tasks and the results mentioned below.
> - “Consider comparing against online learned GPCNs”
>   - Thank you for the suggestion. We added results that show BayesPCN outperforming online learned GPCN in Table 1 of the new paper draft.
> - “Make Figure 1 more clear by adding pointers and removing the fourth line”
>   - We have added the row labels but kept the fourth row to showcase the diversity of images written into memory.

---

> > ### Comment · Reviewer_HskX · 2022-08-08
> > **Small increase in score**
> >
> > Thank you for the updated paper. After reading the updated version, some of my concerns are addressed. However, I'm still puzzled by the contribution. I'm raising my score to borderline accept.

---

> > > ### Author Response · Authors · 2022-08-09
> > > **Response to Reviewer 4**
> > >
> > > Thank you for the response and for raising your score.
> > >
> > > We believe that the core contribution of BayesPCN to the associative memory community is
> > > 1. Being the first parametric associative memory to continually learn hundreds of i.i.d. high-dimensional observations while maintaining recall performance comparable to the state-of-the-art offline learned associative memory (GPCN).
> > > 2. Being the first hierarchical memory to support controlled forgetting that, as a bonus, recovers the empty memory state on its repeated application.
> > > 3. Connecting the predictive coding architecture to probabilistic continual learning (specifically the sequential imputation framework mentioned in section 4.2 https://www.tandfonline.com/doi/abs/10.1080/01621459.1994.10476469) and showing that this leads to a continually learnable associative memory.
> > >
> > > Please let us know if you have further questions so we can do our best to clarify them.

---

### Official Review · Reviewer_yM5G · 2022-07-11

**Rating:** 5
**Confidence:** 3
**Soundness:** 3 good
**Presentation:** 3 good
**Contribution:** 3 good

**Summary:**

This paper extends upon predictive coding neural networks by making the weights Bayesian and using this relationship to make connections to associative memory models within a deep learning framework more broadly. The model is also able to forget over time.

**Questions:**

See above.

**Limitations:**

No I have requested that they add a clear limitations section.

**Strengths And Weaknesses:**

Fig. 1 add labels to the rows and columns rather than putting everything in the caption.

I think you should cite the paper http://proceedings.mlr.press/v80/marino18a/marino18a.pdf because of its relations to predictive coding. Discussing relations between your model and VAEs could be interesting in general.

The Kanerva Machine should be added as a benchmark? Also a standard VAE as per the results of: https://www.pnas.org/doi/abs/10.1073/pnas.2005013117 . I would also cite the follow up to the Kanerva Machine: https://proceedings.neurips.cc/paper/2018/file/6e4243f5511fd6ef0f03e9f386d54403-Paper.pdf

Line 119 difference between hetero and auto associative is confusing. Line 188 talking about SDM as non hierarchical -- this descriptor should apply to the other models mentioned too like MCHNs. Another sentence on the differences between your model and the Kanerva Machine could be helpful.

The images in App. H generated from the BayesPCN are not impressive. Some commentary on this as a limitation in the main text should be present. A limitations section more broadly would be good.

Can you control for the continual learning abilities? E.g. can you learn offline with BayesPCN and see if it outperforms the GPCN?

I'd like to see more experimental results and examples where BayesPCN performs better than existing approaches. You also note that MHN does worse for more heteroassociative tasks but dont show this in Table 1. Also what heteroassociative approach are you using here? The correct one is probably to concatenate the key and value into one large vector as is done in https://arxiv.org/abs/1606.01164

Regarding forgetting it would be useful to know the memory capacity and performance and potentially benchmarking in general against a method like: https://proceedings.mlr.press/v162/sharma22b/sharma22b.pdf

Where does eq. 33 come from?

---

> ### Author Response · Authors · 2022-08-02
> **Response to Reviewer 3**
>
> We would like to thank the reviewer for their time and feedback. We address the points individually and kindly ask the reviewer to increase his/her score based on our response.
>
> - “Fig. 1 add labels to the rows and columns rather than putting everything in the caption”
>   - We have added the row labels.
> - “Discuss connections with iterative amortized inference and VAEs”
>   - While iterative amortized inference is closely connected to predictive coding in that it iteratively fits the variational distribution parameter of the latent variable, we did not include it because it is not an associative memory, not explicitly a predictive coding model, and due to the page limit. The same line of reasoning goes for VAEs. However, we cited Hybrid Predictive Coding while discussing future work in the conclusion, a framework that employs both iterative and amortized inference.
> - “Use the Kanerva machine and VAEs as benchmarks
>   - While the comparison could be interesting, we did not include the Kanerva machine as the benchmark because it is not explicitly an associative memory and employs meta-learning. We also did not evaluate against VAE for similar reasons.
> - "Cite the followup paper to the Kanerva machine"
>   - We have done so in the new draft.
> - “Difference between hetero and auto associative is confusing, mention that non-hierarchical memories are non-hierarchical, add a sentence on the difference between BayesPCN and the Kanerva machine”
>   - We have done so in the new draft.
> - “Mention in the main text the poor model generation quality as a limitation”
>   - We did not include the poor model generation quality in the main text as a limitation because our paper is foremost an associative memory paper that continually learns via probabilistic principles.
> - “Can you control for the continual learning abilities (learn offline with BayesPCN and see if it outperforms GPCN)”
>   - This is an interesting idea and we will investigate this in future work.
> - “More experimental results where BayesPCN outperforms existing approaches”
>   - We added results that show BayesPCN outperforming online learned GPCN in Table 1 of the new paper draft.
> - “Show MHN performing worse for high query noise heteroassociative tasks”
>   - We included this result in Table 3, Appendix G.
> - “What heteroassociative approach are you using for MHN”
>   - We concatenated the key and value vectors into one large vector as you hypothesized.
> - “It would be useful to know the memory capacity and consider benchmarking against hetero-association via scaffolding paper”
>   - This is an interesting idea and we will investigate this in future work.
> - “Where does equation 33 come from”
>   - We can arrive at equation 33 by applying the chain rule and wrapping the numerator and the denominator of the fraction with exp and log functions.
> - "Add a limitation section"
>   - We did not add an explicit limitation section in this draft due to space constraints but adjusted the future work section to better highlight the limitations. We will do this in the camera-ready version of the paper.

---

> > ### Comment · Reviewer_yM5G · 2022-08-08
> > **Response to Authors**
> >
> > > While the comparison could be interesting, we did not include the Kanerva machine as the benchmark because it is not explicitly an associative memory and employs meta-learning. We also did not evaluate against VAE for similar reasons.
> >
> > Kanerva machines are based on SDM which is explicitly an associative memory so I am not sure what you are saying here. The paper I cite for VAEs also shows that they are an associative memory. So I really do think you need these benchmarks and am discouraged by your dismissal of them as useful baselines. For this reason I won't be raising my score.
> >
> > > We concatenated the key and value vectors into one large vector as you hypothesized.
> >
> > Please make this clearer in the main text if you have not already done so.
> >
> > Thanks for your other helpful replies.

---

> > > ### Author Response · Authors · 2022-08-08
> > > **Response to Reviewer 3**
> > >
> > > Thank you for your time and response.
> > >
> > > It appears that there was a slight misunderstanding on our part pertaining to your comment on VAE as associative memory, and we apologize. The paper you cited (https://www.pnas.org/doi/abs/10.1073/pnas.2005013117) employs deep autoencoders as associative memories, not a variational autoencoders. We did not include those results because the GPCN paper shows their model outperforms deep autoencoders at associative recall by a wide margin (Figure 3 and 8 of https://proceedings.neurips.cc/paper/2021/file/1fb36c4ccf88f7e67ead155496f02338-Paper.pdf). If you feel that it is still necessary, we will include the autoencoder result in the camera-ready version of the paper.
> > >
> > > As per Kanerva machine, while its memory module was inspired by SDM, it is foremost a generative model and to our knowledge has never applied in an associative recall settings in spite of the model containing an associative memory within it. In addition, unlike MHN/GPCN/BayesPCN/overparameterized autoencoders, Kanerva machine must be meta-trained over a large amount of data (ex. the entire CIFAR or Omniglot datasets) to demonstrate its impressive generative capabilities. We therefore thought it was not an apples to apples comparison to contrast GPCN/BayesPCN's recall performance after training on at most thousand observations to a model that trains on the entire dataset.
> > >
> > > Lastly, we have made the fact that MHN employs a gradient descent based energy minimization similar to BayesPCN for hetero-associative recall clearer in the appendix.

---

### Official Review · Reviewer_QHLK · 2022-07-11

**Rating:** 5
**Confidence:** 3
**Soundness:** 2 fair
**Presentation:** 2 fair
**Contribution:** 2 fair

**Summary:**

This paper proposes the BayesPCN, an associative memory model that is trained with predictive coding (in contrast to Hebbian learning employed by Hopfield networks). It seems to be an extension of a similar recent work, the generative predictive coding network (GPCN). Rather than using the GPCN's offline write mechanism, this work alters the weights W by Bayesian inference, enabling continual learning in the network. In tandem they propose a forgetting mechanism that incorporates the Bayesian prior. They then compare recall errors for CIFAR10 and Tiny ImageNet across the GPCN, BayesPCN, and the modern Hopfield network (MHN).

**Questions:**

* What are the details of the MHN models used in the experiments for Table 1? What is the size of the MHN? What hyperparameters were searched over?
* Which hyperparameters were best for each model in Table 1? For example, was the forgetting \beta set to 0.0 in these experiments? These hyperparameters could be included in a Supplementary document.
* Lines 242-245 claim that BayesPCN recall quality deterioriated relatively evenly across time. Where is the evidence for this?
* In Table 1 and in Figure 4, is the MSE the average across all datapoints thus far in the seqeuence (e.g. sequence length 512 averages 512 datapoints, whereas sequence length 128 averages 128 datapoints)? Or is it just showing the average across the most recent datapoints?
* What is the standard deviation of MSE across seeds? i.e. can you add these to Table 1 & Figure 4? While the checklist answer does not include these because "we are not necessarily claiming that BayesPCN obtains a better score than the baselines", the discussion does assert that "BayesPCN's recall performance is comparable to its offline-learned counterpart even though it is continually learned." Error bars would be useful to evaluate how comparable they are.

Suggestions
* The PDF appears to be rendered as an image, and it is not possible to search the document or highlight text. Please fix.
* In Section 4.2 line 137, the text refers to the particle model described in Section 3. There is no explicit reference to a particle model in Section 3.
* Line 45: is 'combing' supposed to be 'combining'?

**Limitations:**

Limitations of the model are discussed briefly. e.g. that the model can return nonsensical output for a memory query that is too far away, and that there is not a good biologically plausible implementation of the locally conjugate Bayesian weight update.

**Strengths And Weaknesses:**

The introduction and explanation of the model details, such as the write algorithm, are clear enough. Recasting the GPCN as a Bayesian network is interesting, and to my knowledge the proposed forget mechanism is a novel idea. However, the experiments are the weak point of this paper. Many details on experiment parameters are missing, which I list in the questions below. This makes it challenging to interpret some of the findings. Finally, the claims of novelty in the introduction of the paper are not well-supported by the experiments.

In the first claim, the authors emphasize that the model is continually learnable unlike the GPCN. However, they do not show any results for any common continual learning evaluations, such as class-incremental continual learning. Second, they say that "BayesPCN is the first parametric associative memory model to continually learn hundreds of very high-dimensional observations (>10,000 dimensions) while maintaining good recall performance." While my area of expertise is not in associative memory models, it is my understanding that the capacity of modern Hopfield networks is quite large. The authors do not show any experimental evaluations which result in high error for MHNs and relatively low error for the BayesPCN. They only show an example in Figure 3 of the MHN failing with high dropout and masking. By contrast, this is something more thoroughly evaluated by the GPCN paper (see their Figure 9 and Table 1), though even their work does not provide a good summary comparison between the two.

The authors then state that BayesPCN is "the first hierarchical memory which after repeated forgetting recovers its original memory state." While this third claim may be technically true, it does not seem a particularly significant or notable addition to the literature. If I understand correctly, the authors are saying that a dedicated forget mechanism (other than overwriting old memories) does not exist in models like the GPCN or hierarchical temporal memory (HTM) networks. And the authors show in Figure 4 that BayesPCN recall error is actually smaller later in a sequence, suggesting that it recovers the original images quite well. However, why is it notable or useful to have a *hierarchical* memory that can do this? I assume if it were the first associative memory model to do this, the authors would have claimed that instead.

This is not my area of study, so perhaps I am missing some context that would make this paper's significance more apparent. As is, I am rating it a borderline reject.

---

> ### Author Response · Authors · 2022-08-02
> **Response to Reviewer 2**
>
> We would like to thank the reviewer for their time and feedback. We address the points individually and kindly ask the reviewer to increase his/her score based on our response.
>
> - “The paper does not show any results for any common continual learning evaluations, such as class-incremental continual learning”
>   - We believe the term “continual learning” is justified in our case because we are working with associative memories. While our image datasets do have labeled classes, sequentially memorizing a small to moderate number of images (10s-100s) mostly unrelated to each other in a streaming fashion can be seen as continually learning to solve N independent recall tasks where N is the number of images stored in memory.
> - “The authors do not show any experimental evaluation which result in high error for MHNs and lower error for BayesPCN”
>   - We have included the high query noise experiment result in Table 3 and show that BayesPCN’s recall errors are orders of magnitude lower than MHN’s recall errors in this regime.
> - “Why is it notable to have a hierarchical memory that can forget”
>   - All associative memories (of finite size) have an upper limit on the amount of information they can contain (https://arxiv.org/abs/2202.00159). Therefore, some forgetting is necessary if an associative memory is to continually learn. We believe that intentional forgetting mechanism that improves an associative memory’s recall performance for longer sequences is useful and noteworthy for that reason.
> - “What are the sizes and hyperparameters of the MHN models used in the experiments for Table 1”
>   - Since MHN’s weight matrix is a data matrix, its size was always $<d_{features} \times n_{observations}>$.
> - “Which hyperparameters were best for each model in Table 1”
>   - We did not individually show hyperparameter configuration because the best configuration for cells in Table 1 were often not the same models. However, BayesPCN models with wider hidden layers, GELU, and particle count 4 performed best when the memory was not overloaded.
> - “Where is the evidence of BayesPCN recall deteriorating relatively evenly across time”
>   - It can be seen in the forgetting plot (Figure 3), but we will make this more clear in the camera-ready version.
> - “In Table 1 and in Figure 4, is the MSE the average across all datapoints thus far in the sequence or is it across the most recent datapoints”
>   - It is the average across all datapoints thus far in the sequence and not just the most recent datapoints.
> - “What is the standard deviation of MSE across seeds? i.e. can you add these to Table 1 & Figure 4”
>   - We have added Table 2 in the appendix that also displays the standard deviation. We will add standard deviation to Figure 4 in the camera-ready version.
> - "Are there contexts that would make this paper's significance more apparent"
>   - We believe that connecting predictive coding-based, bio-inspired architecture to probabilistic continual learning and showing that this can lead to a continually learnable associative memory is another theoretical novelty of the paper that was not strongly highlighted in the main text.
>
> We have also added results of BayesPCNs outperforming GPCNs that were naively online-learned in the new paper draft (Table 1). We hope that the newly included experimental results and details help our paper to better support its claims of novelty. Lastly, thank you for the formatting suggestions. They have been incorporated.

---

> > ### Comment · Reviewer_QHLK · 2022-08-02
> > **Questions on broader significance**
> >
> > I thank the authors for their response. Some of my concerns have been addressed. I appreciate the additional data and experimental results in Tables 1-3. In particular, the addition of the naively online GPCN results (as suggested by Reviewer HskX) is useful, and including the BayesPCN with forgetting is critical. The experimental results for a high query noise setting are also appreciated. I am open to increasing my score, but I will wait for continued discussion here before making my final decision.
> >
> > See below for my additional reply.
> >
> > >“Why is it notable to have a hierarchical memory that can forget”
> > All associative memories (of finite size) have an upper limit on the amount of information they can contain (https://arxiv.org/abs/2202.00159). Therefore, some forgetting is necessary if an associative memory is to continually learn. We believe that intentional forgetting mechanism that improves an associative memory’s recall performance for longer sequences is useful and noteworthy for that reason.
> >
> > Yes, it is clear to me why forgetting is useful for a capacity-limited memory, especially in a continual learning setting. I believe the authors missed the point of my question, which is whether there is a special motivation for a forget mechanism in a *hierarchical* memory system, rather than a non-hierarchical memory. That is, after all, what is emphasized on lines 34-35 of the introduction.
> >
> > > "Are there contexts that would make this paper's significance more apparent"
> > We believe that connecting predictive coding-based, bio-inspired architecture to probabilistic continual learning and showing that this can lead to a continually learnable associative memory is another theoretical novelty of the paper that was not strongly highlighted in the main text.
> >
> > I will grant that this does seem somewhat novel, albeit of limited significance. However this point does not seem to be highlighted in the revision.

---

> > > ### Author Response · Authors · 2022-08-03
> > > **Response to Reviewer 2**
> > >
> > > Thank you for your quick response and for being open to increasing the final score.
> > >
> > > As previously mentioned, forgetting is valuable when continually learning associative memories. The main significance of our forgetting mechanism is simply that it provides a way to continual learn hierarchical memories, not just non-hierarchical memories. To our knowledge, only hierarchical memories like GPCN and BayesPCN were able to perform recall when the input query is severely corrupted (ex. 75% masking noise in Table 3), and there is value in being able to continually learn such models. It is also not obvious how to intentionally forget information stored in deep neural network weights in a controlled manner that works.
> > >
> > > We will better accentuate that point and our contribution of connecting a predictive coding architecture to probabilistic continual learning in the camera-ready version of the paper.
> > >
> > > Please let us know if you have unresolved questions or concerns that may affect your decision to increase the final score so we can do our best to address them.

---

> ### Comment · Reviewer_QHLK · 2022-08-08
> **Increased score after author rebuttal and revisions**
>
> I have increased my score from a borderline reject to a borderline accept. This was mostly due to the additional experiments and details that have been included in the revision. The data now better support their claims of novelty. However, I am not convinced that the contribution to the field is significant, and reading the other reviews and rebuttals have not changed that opinion. Therefore this paper remains borderline for me.

---

### Official Review · Reviewer_HvfQ · 2022-07-15

**Rating:** 4
**Confidence:** 5
**Ethics Flag:** Yes
**Soundness:** 3 good
**Presentation:** 3 good
**Contribution:** 3 good

**Summary:**

In this work, the authors have proposed Bayes predictive coding network (PCN), a model inspired by predicting coding literature from Rao and Ballard. They offer the inclusion of hierarchical associative memory, capable of performing continual one-short memory writes without the need for meta-learning heuristics. This mechanism also helps the model gradually forget partial least important information to free memory. Experiments are performed on cifar and tiny-imagenet to show the efficiency of the proposed method.

**Questions:**

* Can authors comment on how the current approach compares/differs against, the neural coding network [1-3] which is proposed to be a stable predicting coding network that works on a wide variety of networks.
* Ablation study with varied value of Sigma_w and Sigma_x is required, current approach uses Sigma_w = 1 and Sigma_x = 0.01
* Can authors report results with deeper BayesSPCN
* How are baseline models trained and did the authors conduct hyperparameter optimizations of those? Can authors report all these ranges?
* Hyper-parameter optimization- No detail about hyper-parameter optimization, authors should provide reasoning how these parameters were chosen.
* Statistical significance: - Experiments should be conducted for K trials and average performance and the standard error should be reported.
* Can authors comment on the robustness and stability of the given model?
* Can authors provide computational overhead using temporal weights both during train and inference?

* [1] https://ojs.aaai.org/index.php/AAAI/article/view/4389
* [2] https://www.nature.com/articles/s41467-022-29632-7
* [3] https://arxiv.org/pdf/1905.10696.pdf

**Ethics Review Area:**

["Discrimination / Bias / Fairness Concerns", "Inadequate Data and Algorithm Evaluation", "Inappropriate Potential Applications & Impact  (e.g., human rights concerns)"]

**Limitations:**

Authors have failed to include Broader Impact and Societal or negative impact sections. Please refer to the NeurIPS deadline, before submitting your manuscripts.

**Strengths And Weaknesses:**

# Strengths
* Well-written paper. However, writing can be improved. Since this paper can be easily understood researchers working on predicting coding and bio-inspired approaches. But for people outside the field, it's difficult to understand.
* Incremental approach
* Model section is well defined.

# Weakness
* Experimental section needs more details
* Related work is missing some citations
* Ablation study is missing

---

> ### Author Response · Authors · 2022-08-02
> **Response to Reviewer 1**
>
> We would like to thank the reviewer for their time and feedback. We address the points individually and kindly ask the reviewer to increase his/her score based on our response.
>
> - “Can authors comment on how the current approach compares/differs against the neural coding network”
>   - Thank you for letting us know of the related work, we have cited them in our paper. We also mentioned that S-NCN, while not explicitly used as an associative memory, continually learns by task-dependent activation sparsity while BayesPCN learns by incorporating uncertainty over the synaptic weights.
> - “Ablation study with varied value of $\sigma_w$ and $\sigma_x$ is required”
>   - We have run preliminary experiments on this but the result’s takeaways were unclear. We will further investigate this and put it in the camera-ready version of the paper.
> - “Can authors report results with deeper BayesPCN”
>   - We will do this in the camera-ready version of the paper.
> - “How are baseline models trained and did the authors conduct hyperparameter optimizations of those”
>   - These details were in the experiment setup and the appendix, but we further clarified them in the new submission. We searched over the same hyperparameters for GPCN that we did for BayesPCN aside from the particle count. We did not hyperparameter tune MHN but used the inverse softmax temperature parameter of 10,000 (equal to setting $\sigma_x=0.01$ that we used for GPCN / BayesPCN based on the perspective in Appendix E and is ideal for recall), which in practice causes softmax to behave like a max function and is ideal for moderate noise memory recall tasks.
> - “Authors should provide reasoning how hyperparameters were chosen”
>   - We set the hyperparameters and their search ranges based on whether they yielded reasonable results on sample runs of GPCN/BayesPCN.
> - “Statistical significance: Perform K trials of the experiments and report the mean and standard error”
>   - Table 1 results were originally the MSE mean across 3 random seeds and we left out the standard deviation due to page limit constraints. In the new submission, we provide both the mean and standard deviation of Table 1 results in the appendix (Table 2).
> - “Can authors comment on the robustness and stability of the model”
>   - Recall performance was largely stable across multiple seeds but we found that both GPCN and BayesPCN can be sensitive to hyperparameters like activation gradient descent learning rate, optimizer, and the hidden neuron value initialization. We specifically found that setting the learning rate too high or using SGD instead Adam can cause the activation gradient descent to diverge.
> - “Can authors provide computational overhead using temporal weights both during train and inference”
>   - We will include this in the camera-ready version of the paper since we did not record it during our experiment. However, online training took much less compute than offline training and obtaining test time results was more computationally intensive than obtaining train time results since we are running 30 ICM iterations (inference on one datapoint is roughly equal in compute to storing 30 data points). While we can get away with lower ICM iterations, we made this choice to obtain the best scores and to ensure that ICM converges, similar to the GPCN paper.
> - “Authors have failed to include Broader Impact and Societal or negative impact sections”
>   - We did not include the broader / societal / negative impact section because we believe there is no particular societal harm that continually learnable associative memories will cause apart from those that always apply to continually learned artificial intelligence agents and we were near the page limit. We will add this to the camera-ready version of the paper.

---

> ### Author Response · Authors · 2022-08-08
> **Response to Reviewer 1 Update**
>
> We have reported the moderate noise associative recall results for different values of $\sigma_W,\sigma_x$ as well as deeper BayesPCN models in Table 4 and Table 5 respectively.

---

> ### Author Response · Authors · 2022-08-09
> **Discussion Ends Today**
>
> Dear Reviewer,
>
> We wanted to inquire whether you were satisfied with our responses below. If not, please let us know so we can provide more details or address your remaining concerns before 1 PM PDT today.
>
> We thank you again for your time!

---

### Review · Ethics_Reviewer_cg85 · 2022-07-27

**Recommendation:** None.

**Ethics Review:**

The paper uses data that contains an MIT licence, and the authors state they are releasing the code under this licence.

---

### Meta-Review · Area_Chair_HZPW · 2022-08-26

**Recommendation:** Accept
**Confidence:** Certain

**Metareview:**

This paper proposes a bayesian extension to predictive coding neural networks, serving as a form of associative memory. The response from reviewers was lukewarm, but almost unanimously erred on the side of acceptance. The only review arguing for borderline rejection was not as substantial as I would have liked, and did not reply to the authors after their rebuttal. Ultimately, I would have preferred to have a clear champion amongst the other reviewers, arguing more strongly for the paper, but having read the discussion I will take the tepid approval as a sign that the reviewers recommend acceptance, and recommend that the paper be included in the proceedings.

**Award:**

No

---

### Decision · Program_Chairs · 2022-09-14

Accept